☮ | **Open Peer Review** | Applied and Industrial Microbiology | Research Article

# Understanding brewing trait inheritance in *de novo* Lager yeast hybrids

Vasni Zavaleta,[1,2] Laura Pérez-Través,[3] Luis A. Saona,[1,4] Carlos A. Villarroel,[2,5] Amparo Querol,[3] Francisco A. Cubillos[1,2,4]

**ABSTRACT** Hybridization between *Saccharomyces cerevisiae* and *Saccharomyces eubayanus* resulted in the emergence of *S. pastorianus*, a crucial yeast for lager fermentation. However, our understanding of hybridization success and hybrid vigor between these two species remains limited due to the scarcity of *S. eubayanus* parental strains. Here, we explore hybridization success and the impact of hybridization on fermentation performance and volatile compound profiles in newly formed lager hybrids. By selecting parental candidates spanning a diverse array of lineages from both species, we reveal that the Beer and PB-2 lineages exhibit high rates of hybridization success in *S. cerevisiae* and *S. eubayanus*, respectively. Polyploid hybrids were generated through a spontaneous diploid hybridization technique (rare-mating), revealing a prevalence of triploids and diploids over tetraploids. Despite the absence of heterosis in fermentative capacity, hybrids displayed phenotypic variability, notably influenced by maltotriose consumption. Interestingly, ploidy levels did not significantly correlate with fermentative capacity, although triploids exhibited greater phenotypic variability. The *S. cerevisiae* parental lineages primarily influenced volatile compound profiles, with significant differences in aroma production. Interestingly, hybrids emerging from the Beer *S. cerevisiae* parental lineages exhibited a volatile compound profile resembling the corresponding *S. eubayanus* parent. This pattern may result from the dominant inheritance of the *S. eubayanus* aroma profile, as suggested by the over-expression of genes related to alcohol metabolism and acetate synthesis in hybrids including the Beer *S. cerevisiae* lineage. Our findings suggest complex interactions between parental lineages and hybridization outcomes, highlighting the potential for creating yeasts with distinct brewing traits through hybridization strategies.

**IMPORTANCE** Our study investigates the principles of lager yeast hybridization between *Saccharomyces cerevisiae* and *Saccharomyces eubayanus*. This process gave rise to the lager yeast *Saccharomyces pastorianus*. By examining how these novel hybrids perform during fermentation and the aromas they produce, we uncover the genetic bases of brewing trait inheritance. We successfully generated polyploid hybrids using diverse strains and lineages from both parent species, predominantly triploids and diploids. Although these hybrids did not show improved fermentation capacity, they exhibited varied traits, especially in utilizing maltotriose, a key sugar in brewing. Remarkably, the aroma profiles of these hybrids were primarily influenced by the *S. cerevisiae* parent, with Beer lineage hybrids adopting aroma characteristics from their *S. eubayanus* parent. These insights reveal the complex genetic interactions in hybrid yeasts, opening new possibilities for crafting unique brewing yeasts with desirable traits.

**KEYWORDS** yeast, beer, volatile compounds, lager, hybridization, RNA-seq

**Peer Reviewer** Carla Bautista Rodriguez, Universite Laval, Quebec, Canada

Address correspondence to Francisco A. Cubillos, francisco.cubillos.r@usach.cl.

The authors declare no conflict of interest.

See the funding table on p. 17.

Hybridization, a phenomenon where genomes from two different species merge, is a significant evolutionary force across various kingdoms such as fungi (1),

Plantae (2), and Animalia (3), facilitating rapid adaptive evolution. This process, more frequent among closely related sympatric species with incomplete prezygotic isolation, increases the genetic diversity of a population (4). Hybrids can exhibit traits that are not simply intermediate between their parents. In some cases, hybrids demonstrate increased fitness, a phenomenon known as hybrid vigor or heterosis (1, 5), which can lead to parental replacement by hybrid swarms (4). Notably, changes in the hybridization frequency between sympatric species have been correlated with novel environmental conditions provided by humans. An example of this is the fermentation environment, where high sugar concentration and temperature restrictions facilitated the hybridization of different *Saccharomyces* species, leading to the replacement of the original parental species (6–8).

The *Saccharomyces* genus is an iconic model system to study hybridization (9). Many *Saccharomyces* species have outcrossed, generating a range of hybrids (10). Notably, most *Saccharomyces* hybrids have been isolated from industrial environments, such as those involved in the fermentation of wine and beer, suggesting that hybridization in yeast is an efficient mechanism to thrive in challenging fermentative conditions (11). The best-known example is *Saccharomyces pastorianus*, the yeast responsible for lager-pilsner beer production by fermenting at low temperatures and the most-produced alcoholic beverage globally (12). *S. pastorianus* resulted from the successful interspecies hybridization between *S. cerevisiae* and *S. eubayanus* (13), showcasing hybridization's adaptive benefits. Although *S. cerevisiae* is a well-studied species widely used in wine and beer fermentation (14), *S. eubayanus* was only recently discovered, with its characteristics remaining unknown until 2010 (13). In this way, the *S. pastorianus* hybrid combines the *S. eubayanus* cold tolerance (due to its mitochondrial inheritance) and the superior fermentation kinetics and sugar consumption capacity of *S. cerevisiae (12)*. Throughout the history of lager yeast, different bottleneck events led breweries to retain only two pure-type strains: the "Frohberg" and "Saaz" lineages. This significantly contributed to a decline in the diversity of lager beer yeast (15).

Industrial lager beers are known for their generally plain and homogeneous aroma profiles (16). However, innovative craft beers with unique profiles are increasingly capturing consumers' attention (17, 18). Novel hybrids have played a significant role in boosting the production of desired fruity and floral volatile compounds (VCs), such as higher alcohols, ethyl esters, and fatty acid esters, in beer and other fermented beverages (19–22). Additionally, certain phenolic and spicy VCs, such as 4-vinyl guaiacol (4-VG), which are typically absent in commercial lager strains due to being considered phenolic off-flavors (POF-), are normally produced by wild yeasts such as *S. eubayanus* (POF+) (15, 23, 24). The utilization of novel hybrid strains using these POF+ strains could expand the beer's flavor profile, providing more complex fruity and spicy characteristics (16, 25, 26).

Novel hybrid fermentation traits have been associated with the *de novo* lager's subgenome composition. In this sense, a recent study using only three *de novo* lager yeast hybrids revealed that higher ploidy levels resulted in higher production of distinct volatile compounds and sugar consumption levels (22). These could likely result from genomic interactions and greater expression levels of flavor-active encoding genes, impacting the unique aroma profile in lager yeasts (22, 27). However, our current understanding of the parental subgenomes' contribution, dosage, and ploidy levels concerning fermentative capacity and volatile compound production in *de novo* hybrids during lager fermentation is still limited. In addition, as the exact *S. cerevisiae* and *S. eubayanus* parental genomes of *S. pastorianus* are unavailable, our understanding of the evolutionary history of the lager hybrid is based on the sequence analysis of reference genomes from the parental species (28, 29) or other lager hybrid genomes (30, 31). Consequently, the complex molecular interactions and the mechanisms by which genome interactions impact the fermentation and aroma profiles of *S. cerevisiae* × *S. eubayanus* hybrids remain largely unknown.

In this study, we aimed to deepen our understanding of *S. cerevisiae* and *S. eubayanus* hybridization, specifically focusing on creating and evaluating a genetically rich set of

hybrids for beer production. Our research explores into these hybrids' fermentative and aroma characteristics, evaluating their efficiency compared with parental strains. Through transcriptome analyses of *de novo* lager hybrids, our goal was to unravel how specific genetic combinations influence these aroma profiles. This research contributes to our knowledge of hybridization inheritance patterns in lager yeast and explores new avenues for enhancing yeast strains in the biotechnology industry.

## MATERIALS AND METHODS

### Yeast strains

Ten diploid *S. eubayanus* strains representative of different genetic lineages isolated from native Chilean forests were chosen (29) (Table S1). All these strains previously exhibited the highest fermentative capacities within each clade (29). In addition, 20 diploid *S. cerevisiae* strains isolated from different anthropogenic niches were included in this study (28). The commercial lager strain *S. pastorianus* W34/70 was used as a fermentation control.

### Wort fermentations

Depending on the specific experiment, fermentations were conducted in 10 mL and 50 mL volumes, using 12 °Plato (°P) beer wort. The wort was oxygenated to 15 mg $L^{-1}$ and supplemented with 0.3 ppm $Zn^{2+}$ (as $ZnCl_2$) at 12°C as previously described (32). Briefly, a pre-inoculum was prepared overnight in 5 mL of 6 °P malt extract (Maltexco, Chile) wort at 20°C, which was then used to inoculate 50 mL culture in 12 °P malt extract under the same previous conditions. We inoculated 50 mL of 12 °P malt extract with $9 \times 10^8$ cells for fermentations. The micro-fermentations were conducted for 14 days at 12°C, and the $CO_2$ production was recorded daily by weighing the bottles. Residual sugars and ethanol production were measured using high performance liquid chromatography (HPLC). For this purpose, 20 µL of each sample filtered through 0.22-µm syringe filters was injected into Shimadzu Prominence HPLC equipment (Shimadzu, USA) and eluted on an Aminex HPX87H column (Bio-Rad, USA) using 5 mM $H_2SO_4$ as mobile phase and acetonitrile 4 mL/L at a flow rate of 0.5 mL/min.

### Generation of polyploid hybrids by rare mating

Polyploid hybrids were generated using the rare mating procedure as previously described (20). We selected natural auxotroph variants of *S. cerevisiae* and *S. eubayanus* by plating overnight cultures on two minimal plates containing 2% glucose and 0.17% YNB without amino acids and $(NH_4)_2SO_4$ (Difco, France). *S. eubayanus* tryptophan auxotrophic variants were selected on minimal media supplemented with 0.05% (wt/vol) 5-fluoroanthranilic acid (5-FAA) (Sigma-Aldrich, USA) and an amino acid stock previously reported (33). *S. cerevisiae* lysine auxotrophs variants were obtained on minimal media plates supplemented with 0.1% wt/vol α-aminoadipic acid (α-AA) (Alfa Aesar, USA) and 30 mg/L lysine (34). Separate overnight cultures in YPD at 25°C were combined, centrifuged, and incubated for 7 days at 12°C in fresh YPD. Potential hybrids were plated on a defined minimal medium consisting of 0.17% YNB without amino acids (Difco, France), 2% glucose, and 2% agar and incubated at 12°C for 7–10 days. Hybrid strains were confirmed by *Hae*III (NEB, USA) digestion of the ITS amplicon obtained using ITS1 and ITS4 primers (22, 35).

As we plated combined cultures multiple times to procure hybrid colonies, we determined the success rate by calculating the frequency of successful attempts. Each attempt encompassed plating the combined culture on batches comprising 10 minimal media plates. A positive attempt was defined as identifying at least one confirmed hybrid within a specific batch of plates.

## Hybrid's genetic stabilization and ploidy level determination

The hybrid strains' genetic stabilization was performed as described by Lairón-Peris et al. (36) and Pérez et al. (20), with minor modifications. Briefly, 20 mL of 12 °P beer wort was inoculated with each hybrid strain for 7 days at 12℃. Following this, 50 µL was used to inoculate fresh medium under the same conditions. This cell transfer process was repeated seven times, corresponding to approximately 42 generations, approximately six generations per transfer passage (37). After the stabilization, the ploidy level of each hybrid was determined using flow cytometry as described by Nespolo et al. (29). A 5 mL overnight culture of each hybrid in YPD medium was prepared, then pelleted, and resuspended in 2 mL of water. Subsequently, 1 mL of the suspension was mixed with 2.3 mL of cold absolute ethanol and stored at 4℃ for 24 h for fixation. The cells were then pelleted and resuspended in 1 mL of 50 mM sodium citrate buffer (pH 7). This step was repeated, and $1 \times 10^7$ cells contained in 100 µL of 50 mM sodium citrate buffer were treated with 1 µL of 100 mg/mL RNase A (Roche, Suiza) and incubated for 2 h at 37℃ to remove RNA. Finally, 350 µL of the labeling solution containing 50 µg/mL propidium iodide (Sigma-Aldrich, USA), 50 mM sodium citrate, pH7, was added, followed by a 40-min incubation in darkness at room temperature. Samples were analyzed using a FACSCanto II cytometer (Becton Dickinson, USA), and around 150.000 single-labeled cells were used for ploidy determination.

## Microculture and temperature tolerance phenotyping

Cells were pre-cultivated at 20℃ without agitation for 48 h in 96-well plates containing 200 µL YPD (1% yeast extract, 2% peptone, and 2% glucose). A volume of 10 µL of pre-inoculum was used to inoculate a new 96-well plate containing 200 µL of YNB 0.67% (Difco, France) supplemented with the following carbon sources: 2% maltotriose (Sigma-Aldrich), 2% maltose (SRL, India), 20% maltose, 2% maltose with 6%, 8%, or 12% (vol/vol) ethanol to an optical density (OD600) of 0.03–0.1. Additionally, the strains were assessed in complex media such as YPD, 12, and 20 °P wort. The optical density (OD) for each well was measured at 620 nm every 30 min for 96 h. The average area under the growth curve (AUC) from triplicates for each strain was calculated using the R-based tool Growthcurver v 0.3.1 (38). Values were normalized between 0 and 1, representing the lowest and highest growth values under a specific condition.

A 10× serial dilution assay was carried out to assess yeast growth across a broad temperature range. For this, overnight YPD cultures of each strain were serially diluted, and 4 µL of each dilution was transferred to YPD plates. Inoculated plates were incubated at 4, 12, 20, 25, 30, and 37℃. Plates incubated at 25, 30, and 37℃ were photographed on the third day. Plates incubated at 20, 12, and 4℃ were photographed on the fourth, eighth, and 12th day, respectively.

## Volatile compounds quantification

Volatile compounds in both the hybrids and parental strains were analyzed using headspace solid phase microextraction (HS-SPME) with a 100 µm polydimethylsiloxane (PDMS) fiber (Supelco, Sigma-Aldrich, Spain). The analysis was conducted using a TRACE GC Ultra gas chromatograph equipped with a flame ionization detector (FID) and a TriPlus RSH autosampler (Thermo Fisher Scientific, Waltham, MA). Samples derived from 14-day beer wort fermentations at 12℃ were prepared by mixing 5 mL of the sample with an equal volume of NaCl saline solution (75 g of NaCl in 247.5 mL mQ-water) in a 20 mL glass vial. Each sample was incubated at 40℃ for 30 min before injection. The PDMS fiber was exposed to the vial's headspace for 15 min, followed by a 5-min desorption at 250℃ in a spitless mode in the gas chromatography (GC)-injection port. Helium was used as the carrier gas at a flow rate of 1 mL/min. An Agilent HP INNOWax capillary column (30 m × 0.25 m), coated with a 0.25-µm layer of cross-linked polyethylene glycol (Agilent Technologies, USA), was employed. The oven temperature program began at 50℃ for 5 min, increased by 1.5 ℃/min to 100℃, then

by 3°C/min to 215°C, and held for 2 min at 215°C. The detector temperature remained constant at 280°C. Chromatographic signals were recorded using ChromQuest software. Fourteen compounds were identified and quantified based on retention times and calibration curves of corresponding standard volatile compounds. 2-Phenyl ethanol, isobutyl acetate, ethyl acetate, 2-phenylethyl acetate, hexanoic acid, octanoic acid, decanoic acid, ethyl hexanoate, ethyl octanoate, and ethyl decanoate were obtained from Sigma Aldrich (USA), whereas isobutanol, isoamyl alcohol, isoamyl acetate, and ethyl propanoate were purchased from Merck (USA).

## Estimation of best parent heterosis

To evaluate best parent heterosis (BPH), we calculated BPH values as previously described by Geng et al. (39). For this, we used the formula BPH = (F1 – BP)/BP, where F1 represents the value of a specific phenotype in the hybrid, and BP corresponds to the value of the same phenotype in the best parent. For heatmap visualization, the BPH value for a specific trait was normalized relative to the highest absolute value recorded among the hybrids for that trait, establishing a scale from 0 to 1. Values in red (>0 to 1) represented heterosis, whereas zero and negative values (0 up to −1) shown in shades of blue and white, respectively, represented the absence of heterosis.

## RNA-seq

RNA was obtained and processed after 24 h under 12°B beer wort fermentation (as indicated in "Wort fermentations") at 12°C in triplicates as previously described (40). Briefly, RNA from hybrids was extracted using E.Z.N.A. Total RNA Kit 1 (Omega Bio-Tek, USA) and subsequently purified using the RNeasy MinElute Cleanup Kit (Qiagen, Germany). The cDNA libraries were generated using the TruSeq RNA Sample Prep Kit v2 (Illumina, San Diego, CA, USA). Illumina libraries from hybrids were generated using paired-end 150 bp reads on an Illumina Next seq 500 as previously described (41). RNA sequencing data from HB6 and HB41 hybrid strains were mapped against a concatenated *S. cerevisiae* R64-1-1 and *S. eubayanus* genome CL216.1 (42) using STAR (--outSAMmultNmax 1), after which gene counts were obtained using featureCounts (43, 44). Expression count data were imported into R, with gene identifiers updated by parental species-specific mappings (*S. cerevisiae* and *S. eubayanus*). Differential expression was assessed using the DESeq2 package, version 4.1.2 (45), applying a log2 fold change threshold (log2FC) of |1| and an adjusted *P*-value threshold (*P*-adj) of <0.01. Differentially expressed genes (DEGs) were thus identified in all conditions, and DEGs from HB6 (*P*-adj < 0.01 and log2FC > 1) and HB41 (*P*-adj < 0.01 and log2FC < −1) were subsequently extracted for further analysis.

Metabolic pathway enrichment was analyzed using the enrichKEGG and enrichGO tools (46) based on gene identifiers in Entrez format. Analyses were performed separately for each hybrid strain. The results of Kyoto Encyclopedia of Genes and Genomes (KEGG) pathway enrichment and GO terms were visualized using dot and bar graphs.

## Statistical analysis

The data were deemed statistically significant with a *P*-value < 0.05, calculated using one-way ANOVA (analysis of variance) with Tukey's honestly significant difference (HSD) test *post hoc* and the nonparametric Wilcoxon–Mann–Whitney test. These analyses were conducted in R (version 4.3.1) using the AOV and Wilcox.test functions, respectively. Plots and heatmaps were also generated with the ggplot2 (version 3.4.3) and pheatmap (version 1.0.12) packages. The principal component analysis (PCA) was performed using the function prcomp from stats package v. 4.3.1 and the ggfortify v. 0.4.16 and ggplot2 v. 3.4.3 packages for extracting, visualizing, and interpreting the results.

## RESULTS

### High hybridization rate across most *S. cerevisiae* and *S. eubayanus* lineages

To maximize the genetic diversity of *S. cerevisiae* × *S. eubayanus* hybrids, we employed a crossbreeding strategy using yeast strains from distinct genetic lineages of both species. Initially, we aimed to identify parental strains with robust fermentative performance in beer wort at low temperatures. We evaluated 30 diploid strains, including 10 *S. eubayanus* and 20 *S. cerevisiae* strains, representing various lineages. The selection of *S. eubayanus* strains was based on their previously reported high fermentative capacity (29). In agreement with our previous findings, no significant variations in fermentative capacity across *S. eubayanus* were observed ($P$-value > 0.05, one-way ANOVA; Fig. 1A), and none of them exhibited maltotriose consumption (Table S2A). Notably, the fermentative capacity of these strains closely resembled that of the commercial strain W34/70 ($P$-value > 0.05, one-way ANOVA; Fig. 1A). In contrast, *S. cerevisiae* strains showed significant differences in fermentative capacity ($P$-value < 0.05, one-way ANOVA; Fig. 1B). Strains from the Mosaic-Beer (hereinafter referred to as "Beer"), Sake, Bioethanol, and specific strains from the Wine lineages displayed the highest fermentative capacity, with no significant differences compared with W34/70 (Fig. 1B). Remarkably, six strains —Sc_Beer_1 , Sc_Beer_2 , Sc_Wine_1 , Sc_Beer_3 (Beer), Sc_Sake_1 , and Sc_Wine_2— exhibited maltotriose consumption exceeding 60% (Table S2B). Strains from the African Beer, Mexican Agave, French Dairy, and most of the Wine/European lineages showed a lower fermentative capacity than the commercial strain ($P$-value < 0.05, one-way ANOVA; Fig. 1B). Based on the fermentative capacity and sugar consumption profiles, we selected six representative *S. eubayanus* strains from different lineages (Se_PB2_1 (PB-2), Se_PB2_2 (PB-2), SePB1_1 (PB-1), Se_PB3_1 (PB-3), Se_Adm_1 (Admixed), and Se_PB3_2 (PB-3)) and seven *S. cerevisiae* strains (Sc_Wine_1 (Wine), Sc_Beer_1 (Beer), Sc_Beer_2 (Beer), Sc_Sake_1 (Sake), Sc_Wine_2 (Wine), Sc_BEth_1 (Bioethanol), and Sc_BEth_2 (Bioethanol)), all characterized by high fermentative capacity and maltotriose consumption levels in *S. cerevisiae*. These chosen strains represented the genetic starting material to generate *S. cerevisiae* × *S. eubayanus* hybrids.

To generate polyploid hybrids, we employed a rare mating strategy involving crossbreeding complementary amino acid auxotrophic individuals from both species (Fig. 1C). We successfully obtained auxotrophic colonies for both species, except for the *S. cerevisiae* Sc_Wine_1 wine strain. Each auxotrophic strain was subjected to crossbreeding at low temperatures (to facilitate the selection of hybrids containing the *S. eubayanus* mitochondria) with its counterpart species in liquid media. Among the 36 potential cross combinations, 21 produced at least one positive hybrid colony (Fig. 1D). The remaining combinations did not evidence hybrid colonies, likely due to genetic incompatibilities related to mitochondrial and nuclear interactions. A total of 93 hybrid colonies were identified, varying from 1 to 15 hybrids per cross, with a median of 3 hybrids per combination. Remarkably, almost all strains could crossbreed with at least one strain from the opposing species. The beer clade *S. cerevisiae* strain Sc_Beer_2 demonstrated the highest hybridization success rate (defined as the "number of positive attempts/the number of cross-attempts") among *S. cerevisiae* strains at 52.6% (Table S3). In contrast, the *S. cerevisiae* bioethanol Sc_BEth_1 strain did not produce hybrids with any *S. eubayanus* parental strain. Among the *S. eubayanus* strains, Se_PB2_1 (PB-2 lineage) had the highest hybridization success rate at 38.9% (Table S3), whereas Se_PB3_1.1 represented the strain with the highest number of hybrid colonies (31 hybrids after five positive cross-attempts; Fig. 1D). Notably, the cross between Se_PB3_1 × Sc_Beer_1 resulted in the highest number of hybrid colonies from a single cross, producing 15 hybrids. These results highlight the substantial viability of generating inter-species hybrids across most *S. cerevisiae* and *S. eubayanus* lineages.

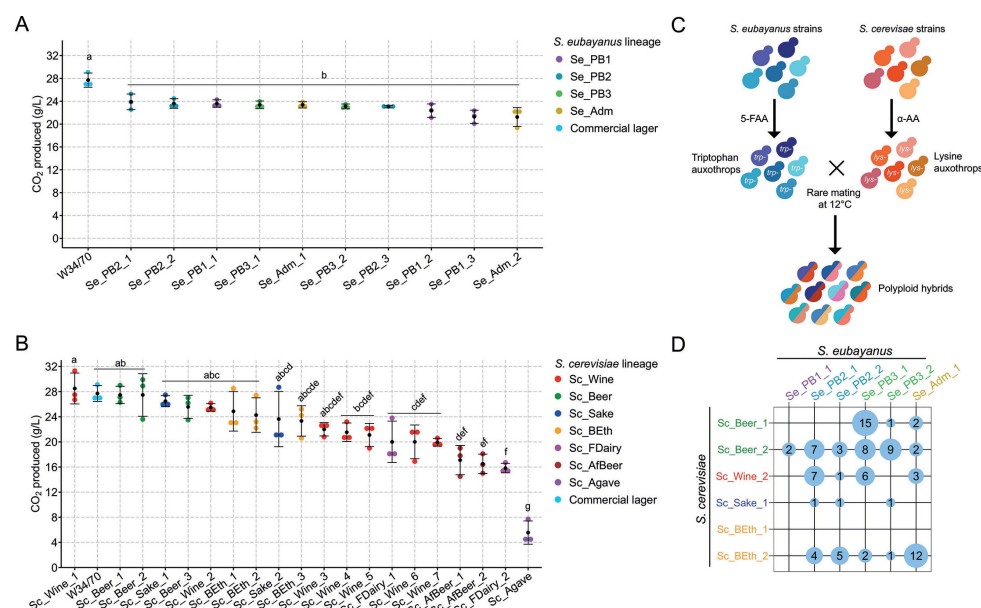

**FIG 1** Hybridization between *S. cerevisiae* and *S. eubayanus* strains. (A) Total $CO_2$ production (g/L) in (A) *S. eubayanus* and (B) *S. cerevisiae* strains under 12 °P wort from a broad range of lineages in both species. Black dots depict mean values between the three replicates. (C) Hybridization strategy to generate *de novo* polyploid lager hybrids using a rare mating approach. Tryptophan (*tyr*−) and lysine (*lys*−) auxotrophs were generated in *S. eubayanus* and *S. cerevisiae* under 5-fluoroanthranilic acid (5-FAA) and α-aminoadipic acid (α-AA), respectively. Subsequently, rare mating at 12°C was performed, and hybrids were selected using minimal media. (D) Hybridization success rate between *S. cerevisiae* and *S. eubayanus* strains. Different letters (from a to g) in panels A and B depict statistically significant differences between strains with a *P*-value < 0.05, one-way ANOVA with Tukey's HSD test *post hoc*.

## The hybrid's fermentative profile is dependent on the *S. cerevisiae* parental lineage

We evaluated the fermentative capacity of 47 hybrids resulting from 21 different crosses, including their parental strains and the commercial strain W34/70 (Table S4). These hybrids exhibited a broad range of $CO_2$ production levels, covering the parental phenotypic space with average values spanning from 19.7 to 35 g/L (Fig. 2A; Table S5). Notably, hybrids sharing the parental *S. cerevisiae* strain Sc_Beer_2 from the beer lineage exhibited the highest $CO_2$ production levels (average 31.58 ± 2.17 g/L, *P*-value < 0.05, Mann–Whitney–Wilcoxon test; Fig. S1A). Building upon this observation, we investigated whether the lineage significantly influenced the hybrids' fermentative capacity. Indeed, hybrids originating from the *S. cerevisiae* beer lineage consistently showed greater $CO_2$ production (31.07 ± 2.21 g/L) compared with hybrids from other lineages (*P*-value < 0.05, Mann–Whitney–Wilcoxon test; Fig. 2B). Conversely, hybrids from the Wine, Bioethanol, and Sake lineages did not exhibit significant differences in $CO_2$ production among themselves (26.64 ± 2.19, 26.73 ± 3.11, and 28.19 ± 0.65 g/L, respectively; *P*-value > 0.05, Mann–Whitney–Wilcoxon test; Fig. 2B). Interestingly, hybrids stemming from distinct *S. eubayanus* Patagonian clades demonstrated comparable levels of $CO_2$ production (*P*-value > 0.05, Mann–Whitney–Wilcoxon test; Fig. 2C; Fig. S1B). Hybrids from the *S. cerevisiae* Beer lineage also exhibited superior maltotriose consumption (56.5% ± 30.38%, *P*-value < 0.05, Mann–Whitney–Wilcoxon test) compared with hybrids from other lineages (7.0% ± 1.90%; 6.62% ± 1.19%; 5.31% ± 1.02 for Sake, Wine, and Bioethanol, respectively; Fig. 2D; Fig. S2; Tables S6 and S7). These results suggest that the fermentation potential of hybrids is predominantly influenced by the *S. cerevisiae* lineage, rather than the *S. eubayanus* parent.

Given that the generation of *S. eubayanus* × *S. cerevisiae* hybrids involved rare mating, the resulting hybrids may present diverse ploidy levels potentially associated with

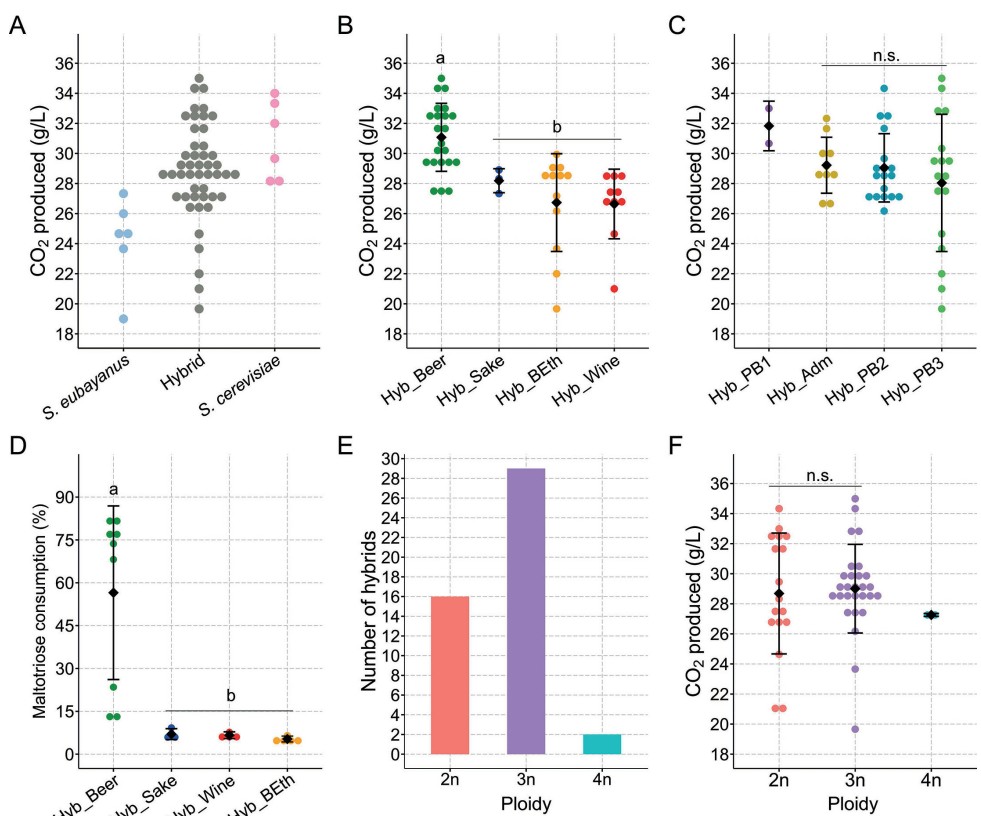

**FIG 2** Fermentative capacity of *S. cerevisiae* × *S. eubayanus* hybrids. (A) Average $CO_2$ production levels in *S. eubayanus* (blue), Hybrids (black), and *S. cerevisiae* (red) strains under 12 °P wort. $CO_2$ production levels in *de novo* lager hybrids depending on the (B) *De novo* hybrids grouped by *S. cerevisiae* parental lineage: Beer (green), Sake (blue), Bioethanol (orange), and Wine (red) lineages and (C) *de novo* hybrids grouped by *S. eubayanus* lineages: PB-1 (purple), PB-2 (light blue), PB-3 (light green), and admixed (gold). (D) Maltotriose consumption levels (%) in *de novo* lager hybrids depending on the *S. cerevisiae* parental lineage: Beer (green), Sake (blue), Bioethanol (orange), and Wine (red) lineages. (E) Ploidy levels determined by FACS in *de novo* lager hybrids after rare mating. (F) $CO_2$ production levels depend on the ploidy level. Mean values are depicted by diamonds. Different letters (a to b) reflect statistically significant differences between strains with a *P*-value < 0.05, Mann–Whitney–Wilcoxon test. n.s. denotes non-significant differences.

brewing-relevant phenotypic traits, such as fermentative capacity. To explore this possibility, we analyzed the ploidy levels of the 47 hybrid strains. The observed ploidy spanned from diploid (2 n) to tetraploid (4 n) (Fig. 2E; Fig. S3; Table S8). Specifically, 16 strains were identified as diploid, 29 as triploid, and two displayed a tetraploid state. These results suggest that the rare mating strategy usually involves a 2n × 1n cross. Next, we explored the relationship between ploidy and the fermentative capacity of the 47 hybrids. Upon analyzing these parameters, we did not detect significant differences in the impact of ploidy on $CO_2$ production (*P*-value > 0.05, Mann–Whitney–Wilcoxon test; Fig. 2F). Similarly, when examining the influence of *S. cerevisiae* and *S. eubayanus* lineages per ploidy level on fermentative capacity, we found no significant differences (*P*-value > 0.05, Mann–Whitney–Wilcoxon test; Fig. S4). In the analysis at the individual parental strain level, we found that the *S. eubayanus* Se_PB2_1 strain exhibited a notable exception, showing higher $CO_2$ production in 3 n hybrids than 2 n hybrids (*P*-value < 0.05, Mann–Whitney–Wilcoxon test; Fig. S4D). However, this corresponded to a particular case not observed in other genetic backgrounds. Altogether, the species and lineage level analysis suggests that the *S. cerevisiae* lineage is the primary determinant of the hybrid's fermentative capacity and that ploidy levels might not influence this trait in the lager's hybrid background.

## S. cerevisiae lineages influence ethanol and osmotic stress tolerance

To evaluate fitness differences among various S. cerevisiae × S. eubayanus hybrids, we chose a single representative hybrid per cross combination (21 hybrids). We subjected them to phenotypic assessments and estimated the AUC under different microculture conditions, including diverse growth temperatures (4°C, 12°C, 20°C, 25°C, 30°C, and 37°C), ethanol tolerance levels (6%, 9%, and 12% [vol/vol]), and carbon sources pertinent to the beer brewing process (2% and 20% maltose, 2% maltotriose, and 12 °P and 20 °P wort). Generally, hybrids originating from the Beer lineage exhibited the highest fitness across the tested conditions compared with other lineages, except under 12% ethanol concentration (Fig. 3A, Hybrid cluster 2; Table S9). Conversely, hybrids including the Wine lineage parental strain demonstrated greater AUC levels under ethanol 12% (Fig. 3A and B, hybrid cluster 1, $P$-value < 0.05, Mann–Whitney–Wilcoxon test), representing the only scenario where hybrids from the Beer lineage did not display the highest fitness. At the same time, S. eubayanus parental strains exhibited the lowest growth fitness values across conditions, indicating that in most instances, hybrids inherited the phenotypic profile of S. cerevisiae (Fig. 3A and C).

Subsequently, we investigated yeast growth under various temperature conditions. Hybrids displayed a wider growth range across temperatures than their parental species (Fig. 3D; see Fig. S5 for the complete set of strains and temperatures: 4°C, 12°C, 20°C, 25°C, 30°C, and 37°C). Notably, at low temperatures (4°C; Fig. S5), hybrids exhibited similar or higher fitness than their S. eubayanus and S. cerevisiae parents, respectively. This pattern persisted at 30°C and 37°C, where hybrids displayed comparable or superior fitness relative to their S. cerevisiae and S. eubayanus parents, respectively (Fig. S5).

## The S. cerevisiae parental lineage predominantly determines the hybrid's volatile compound profile

To explore the novel lager hybrids' aroma profiles, the 21 hybrids previously assessed were selected to assess the production of volatile compounds (VCs) after beer fermentation. These hybrids showed distinct VC profiles compared with their parental strains, clustering separately (Fig. 4; Table S10). We identified four distinct hybrid clusters based on their VC profile. The Hybrid cluster 1 contained uniquely hybrid HB20 (Se_PB3_2 × Sc_Beer_1), aromatically distinctive from other hybrids and characterized by displaying the low levels of most aromas among the strains, except for isobutanol (59.1 ± 1.22 mg/L, sweet, solvent), 2-phenyl ethanol (117.0 ± 3.78 mg/L, roses), and ethyl propanoate (0.77 ± 0.07 mg/L, fruity) (Fig. 4). The Hybrid cluster 2 predominantly steamed from the S. cerevisiae Sc_Beer_2 strain (beer clade). However, all these hybrids exhibited a VC profile similar to their corresponding S. eubayanus parental strains, likely reflecting a recessive Sc_Beer_2 strain VC profile inheritance. These hybrids produced low levels of isobutanol (31.6 ± 2.49 to 44.4 mg/L ± 5.34, sweet, solvent) and higher levels of 2-phenylethyl acetate (1.8 ± 0.16 to 2.8 ± 0.15, roses), ethyl acetate (23.77 ± 1.79 to 33.59 ± 2.36, fruity, sweet), and other alcohols, such as 2-phenyl ethanol (100.9 ± 1.94 to 120.15 ± 3.44, roses) and isoamyl alcohol (226.9 ± 17.56 to 280.5 ± 1.97, whiskey, solvent) (Fig. 4). The Hybrid cluster 3, which involved strains obtained from crosses including the Bioethanol Sc_Beth_2 and Sake Sc_Sake_1 S. cerevisiae parents, was characterized by the presence of higher levels of ethyl decanoate (0.11 ± 0.01 to 0.238 ± 0.05 mg/L, fruity) and medium-chain fatty acids, including octanoic (4.07 ± 0.58 to 7.76 ± 0.46 mg/L) acid and decanoic acid (1.5 ± 0.64 to 3.4 ± 0.39 mg/L) (waxy or fatty flavors) (Fig. 4). The Hybrid cluster 4, primarily emerging from crosses involving the wine Sc_Wine_2 and beer Sc_Beer_1 S. cerevisiae strains, exhibited lower production levels of most VCs (Fig. 4).

Subsequently, we determined the correlation between VC production in the hybrid backgrounds and the corresponding parental lineages in each species. No significant differences were observed across S. eubayanus lineages ($P$-value > 0.05, Mann–Whitney–Wilcoxon test; Fig. S6; Table S11). Conversely, hybrids from different S. cerevisiae lineages exhibited significant differences in most evaluated aromas (Fig. S7; Table S11), particularly the brewing lineage, which displayed a fruity and sweet profile with a significantly

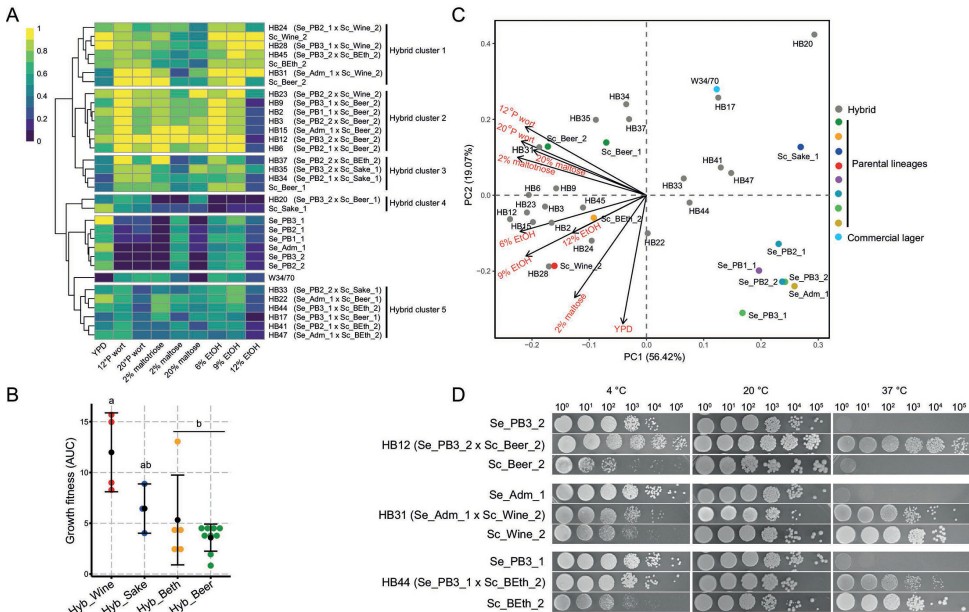

**FIG 3** Phenotypic variability across *de novo* lager hybrids. (A). Heat map depicting the phenotypic diversity in *de novo* lager hybrids obtained from ten assessed conditions. Strains are grouped by hierarchical clustering from AUC data, and names and colors highlight groups of hybrids with similar phenotypes. The heat maps were elaborated based on a 0–1 normalization within each phenotype, with 0 and 1 representing the lowest and highest growth values, respectively. (B) AUC levels for 12% ethanol tolerance in different hybrids depending on the *S. cerevisiae* lineage. Black dots depict mean values. Different letters (a to b) reflect statistically significant differences between strains with a *P*-value < 0.05, Mann–Whitney–Wilcoxon (C) PCA depicting the overall association of hybrids and *S. cerevisiae* or *S. eubayanus* parental lineages (D) Plate spotting assay using 10-fold serial dilution of the HB12, HB31 and HB44 hybrids and its corresponding parental strains grown at different temperatures (4°C–20°C and 37°C).

higher production of ethyl acetate (23,01 ± 8.39), isoamyl alcohol (236.5 ± 28.0), and 2-phenyl ethanol (109.9 ± 7.80) (*P*-value < 0.05, Mann–Whitney–Wilcoxon test; Fig. S7; Table S11). Hybrids emerging from the beer clade exhibited significantly lower production levels of undesired VC, such as octanoic and decanoic acids (2.1 ± 0.93 and 0.65 ± 0.48, respectively) than hybrids from the bioethanol lineage (4.5 ± 0.35, 2.5 ± 0.51, *P*-value > 0.05, Mann–Whitney–Wilcoxon test; Fig. S7). Since *S. eubayanus* and some domesticated yeasts produced the phenolic aroma 4-vinyl guaiacol (4-VG (15, 42), we selected nine hybrid strains from the Beer and Bioethanol lineages to evaluate their production. We detected similar production levels of 4-VG in hybrids and their corresponding parental strains with no significant differences (*P*-value > 0.05, ANOVA; Fig. 4B; Table S12).

To evaluate whether novel hybrids displayed enhanced phenotypic traits compared with their parental strains, we assessed best parent heterosis (BPH) across each VC (Fig. 5). Our analysis revealed 20 hybrids exhibiting BPH for at least one VC, except for the HB28 hybrid (Table S13). Interestingly, hybrids sharing the *S. cerevisiae* parental strain Sc_Beer_2 strain exhibited six of 14 VCs with BPH, demonstrating the high levels of heterosis for VC production in the novel hybrids (Table S13). BPH hierarchical clustering revealed four distinct hybrid clusters, determined mainly by the *S. cerevisiae* lineage (Fig. 5). For example, Hybrid cluster 3 predominantly included hybrids from the beer parent Sc_Beer_2 and displayed higher BPH values for ethyl acetate (0.14 ± 0.15 to 0.68 ± 0.09, fruity, sweet), isobutyl acetate (0.08 ± 0.04 to 0.75 ± 0.04, fruity, banana), and isoamyl acetate (0.39 ± 0.37 to 1.46 ± 0.19, banana; Fig. 5; Table S13). When examining BPH correlation depending on the *S. eubayanus* parental lineage, significant differences were solely found for ethyl propanoate between PB-2 (−0.32 ± 0.18) and the PB-3 (−0.12 ± 0.17) and Admixed lineages (0.09 ± 0.20) (*P*-value < 0.05, Mann–Whitney–Wilcoxon test;

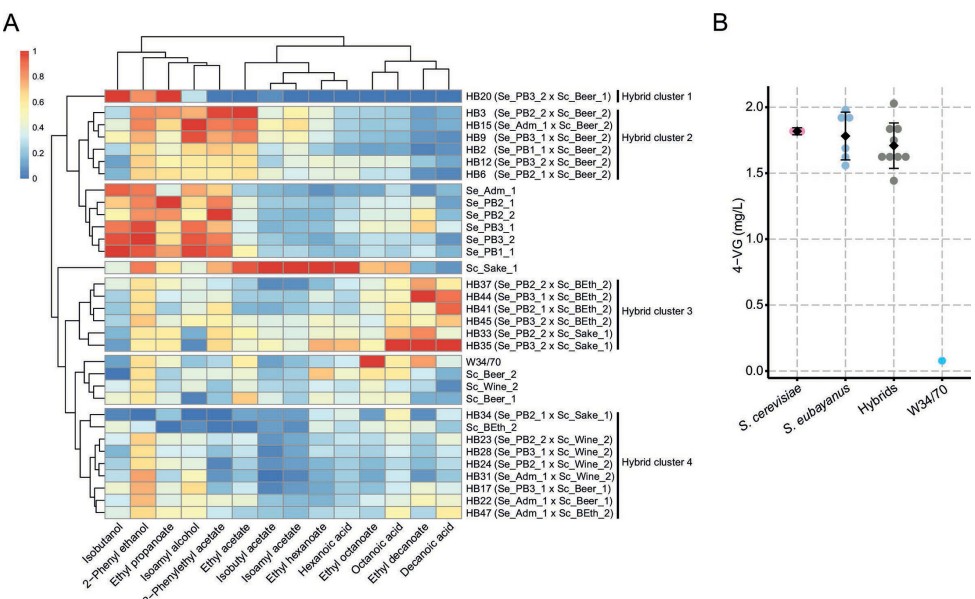

**FIG 4** Volatile compound production profile of *de novo* lager hybrids. Beer wort fermentations were carried out for 14 days at 12°C, and volatile compounds were analyzed by GC-FID at the fermentation endpoint. (A) We used a 0–1 normalization within each phenotype, with 0 and 1 representing the lowest and highest VCs values, respectively. Clusters containing strains with similar aroma profiles are highlighted with colored heatmap sidebars. Hybrids' aroma profiles were classified into four distinctive clusters. (B) 4-VG production of hybrids (gray) from Beer and Bioethanol ancestry are represented along with their *S. eubayanus* (blue) and *S. cerevisiae* (red) parental strains and the commercial strain W34/70. Diamonds depict mean values. No significant differences were detected among hybrids and their parental species (*P*-value > 0.05, ANOVA with Tukey's HSD test *post hoc*). Commercial strain W34/70 (light blue) did not show 4-VG production.

Fig. S8; Table S14). However, in *S. cerevisiae,* we found a significant trend depending on the lineage. In this case, the brewing lineage exhibited significantly greater BPH levels in isoamyl acetate (0.38 ± 0.77) and ethyl propanoate (−0.03 ± 0.21) than Sake (−0.67 ± 0.24) and Wine/European lineages (−0.27 ± 0.24), respectively (*P*-value < 0.05, Mann–Whitney–Wilcoxon test; Fig. S9; Table S14). Similarly, hybrids from the bioethanol lineage displayed high BPH values in ethyl hexanoate (−0.13 ± 0.22), ethyl octanoate (−0.07 ± 0.15), ethyl decanoate (0.18 ± 0.29), hexanoic acid (0.07 ± 0.25), octanoic acid (0.20 ± 0.09), and decanoic acid (1.75 ± 0.55) production, mostly distinguishing themselves from brewing and wine lineage strains (*P*-value < 0.05, Mann–Whitney–Wilcoxon test; Fig. S9; Table S14). These results demonstrate that differences in the hybrid's VC profile are predominantly exerted by the *S. cerevisiae* parental lineage rather than being significantly affected by the *S. eubayanus* lineage.

Finally, exploring the correlation between ploidy and BPH did not detect an overall significant correlation (*P*-value > 0.05, Mann–Whitney–Wilcoxon test; Fig. S10 and S11). However, triploid strains exhibited the greater variance (CV = 0.580), including the highest BPH values compared with diploids (CV = 0.295) and tetraploids (CV = 0.186) (Fig. S10). Since only two tetraploid strains were available, we could not compare the volatile compound production on these hybrids with other ploidy levels. These results suggest an increased BPH phenotypic variability among triploid hybrids compared with diploids.

## Gene expression differences between polyploid hybrids

To understand how gene expression underlies the inheritance of brewing traits, we conducted an RNA-seq analysis after 24 h under beer wort on two *de novo* hybrids characterized by distinct VC profiles exhibiting BPH: HB6 (Se_PB2_1 × Sc_Beer_2) and HB41 (Se_PB2_1 × Sc_BEth_2). By examining mean read counts per gene across

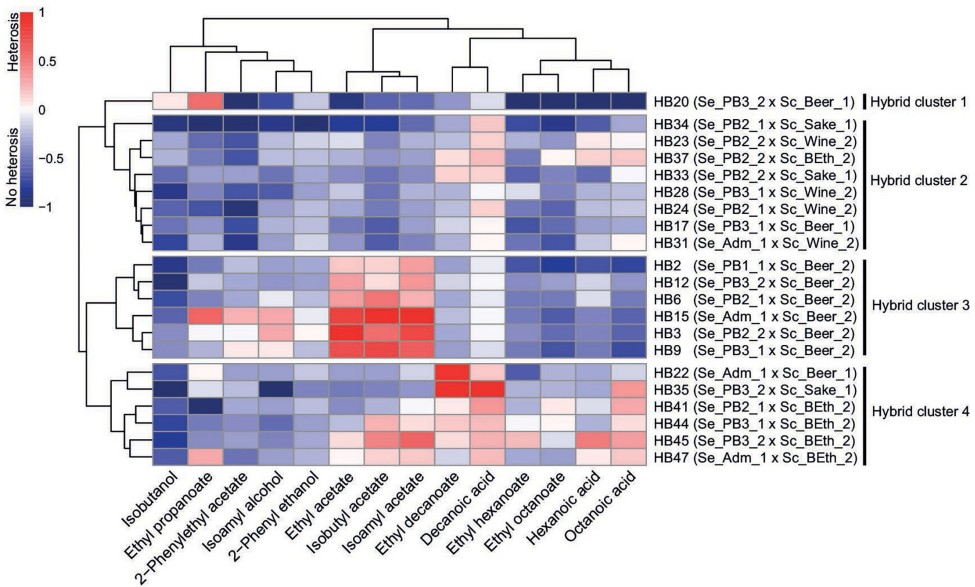

**FIG 5** Best parent heterosis profile based on volatile compound production in *de novo* lager hybrids. Normalized positive (red) and negative (blue) BPH values are depicted on a scale from −1 to 1 relative to the highest absolute BPH value. Hybrids' BPH profiles were classified into four distinctive clusters.

chromosomes, we observed a differential representation of parental genomes in each strain, suggesting that HB6 has a greater representation of *S. cerevisiae* (2 n *S. cerevisiae* × 1 n *S. eubayanus*), whereas HB41 exhibited an opposite pattern (1 n *S. cerevisiae* × 2 n *S. eubayanus*) (Table S15). Subsequently, we compared differential gene expression between hybrids. We found 138 genes upregulated in HB6 and 178 in HB41 (log2 fold change >1; Fig. 6A; Table S15). According to KEGG pathways, the metabolic pathways enriched in each hybrid strain indicated HB6's emphasis on ribosome activity and sugar metabolism, whereas HB41's exhibited over-expression on pathways such as biosynthesis of secondary metabolites and the biosynthesis of amino acids (Fig. 6B). In parallel, according to GO terms, we observed disparities in carbohydrate metabolism between both hybrids. Specifically, HB6 demonstrated upregulation of genes associated with maltose metabolism, including the maltase *MAL12*, the permease *MAL31*, the transcriptional activator *MAL33*, and the low glucose-induced transporter *HXT14* (Fig. 6A). In contrast, HB41 exhibited upregulation of genes primarily linked to glucose metabolism, such as *HXT2*, *HXT4*, and *HXT6* (Fig. 6A). Additionally, HB6 displayed higher gene expression levels in stress tolerance genes, such as *TPS2*, *TSL1*, and *HSP30*. *TPS2* and *TSL1* are implicated in trehalose biosynthesis, whereas *HSP30* contributes to tolerance against ethanol.

Next, we determined the impact of gene expression differences on VC production. For this, we focused on genes and pathways related to HB6 displaying a high production for acetate esters, as representative of other hybrids from the Beer lineage. This analysis highlighted *ADH2* and *ALD6,* which showed increased expression levels in HB6 (Fig. 6A). *ADH2* catalyzes the conversion of ethanol to acetaldehyde, a substrate utilized by *ALD6* to produce acetate (Fig. 6C). Furthermore, we observed upregulation of *ILV6*, a subunit of the acetolactate synthase complex involved in branched-chain amino acid biosynthesis within the mitochondria, serving as precursors for isoamyl alcohol and isoamyl acetate (Fig. 6C). In contrast, HB41, representative of hybrids displaying heterosis in medium-chain fatty acids and their respective ethyl esters showed upregulation in genes associated with fatty acid metabolism such as *POT1*, *ECI1*, *MGA2*, and *IZH4* (Fig. 6A). *POT1* and *ECI1* are peroxisome genes of β-oxidation (Fig. 6C). *MAG2* is involved in the synthesis of unsaturated fatty acids and induces the activity of *IZH4*, a membrane protein with a potential role in sterol metabolism. Likewise, in HB41, we observed upregulation

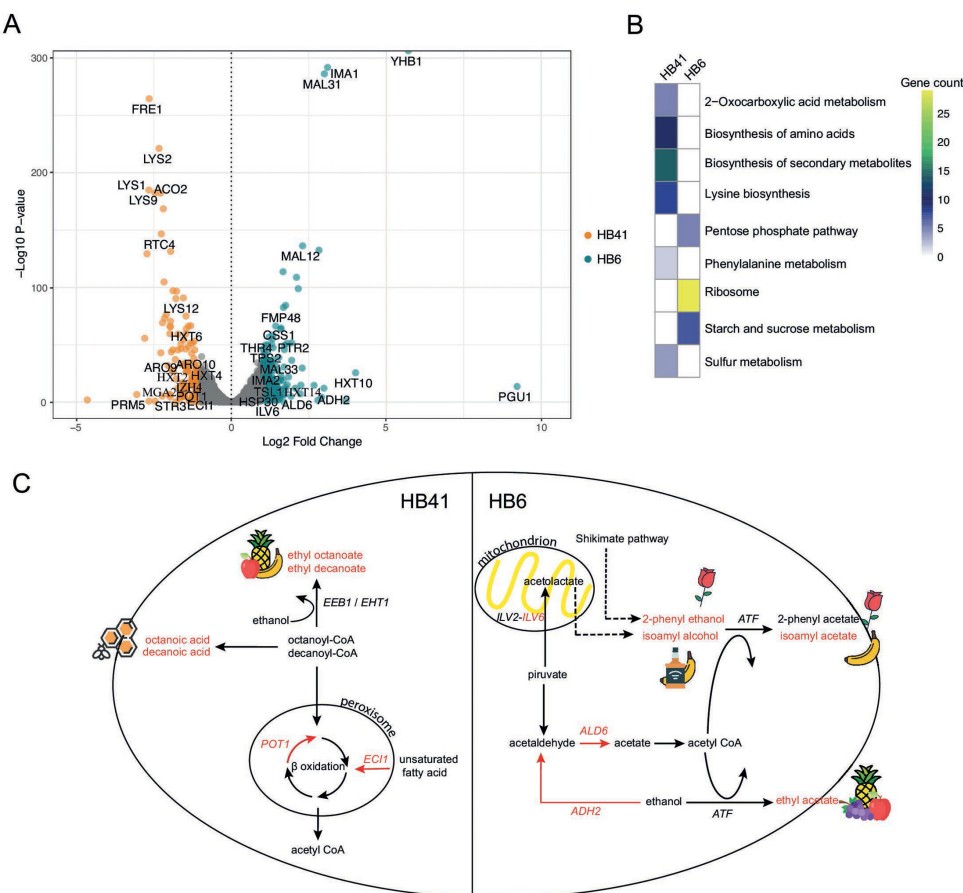

**FIG 6** Transcriptome analysis between HB6 and HB41 hybrid strains. (A) Volcano plot depicting differentially expressed genes (DEGs) and upregulated genes in HB41 (orange) and HB6 (green) hybrid strains. (B) Enriched KEEG pathways in HB6 and HB41 hybrids. Colors depict the number of genes in each category. (C) Metabolic relationship between highly produced volatile compounds and differentially expressed genes. Volatile compounds highly produced in HB6 are displayed on the right panel, whereas those by HB41 are displayed on the left. Red color depicts genes and VCs' greater levels.

of *ARO9* and *ARO10*, genes that participate in the metabolism of aromatic amino acids and their corresponding acetate esters. Although we did not assess the production of aromatic thiols, we detected a higher expression of *STR3* in this strain, a peroxisomal β-lyase implicated in the production of 3-mercaptohexanol (3MH, grapefruit).

## DISCUSSION

Hybridization has played a pivotal role in driving evolution across many lineages with immediate phenotypic consequences through the expression of hybrid vigor (47). One example is *S. pastorianus*, the yeast instrumental in lager fermentation, which arose from the hybridization between *S. cerevisiae* and *S. eubayanus* (8, 27). However, our understanding of the *S. cerevisiae* and *S. eubayanus* hybridization success remains limited, largely due to the need for a wide range of *S. eubayanus* parental strains. Previous studies have only utilized a single *S. eubayanus* parental strain, constraining our understanding of the hybridization process (22, 48–50). The recent identification of diverse *S. eubayanus* lineages in the Patagonian region represents an opportunity to partly elucidate the molecular underpinnings driving the robust hybrid vigor observed in *S. pastorianus* (29, 51).

Polyploid hybrids can be generated through the rare mating technique (20, 52), representing the most successful strategy to generate artificial hybrids in the *Saccharomyces* genus (20, 36, 37). This process exploits yeast strains that could have undergone

loss of heterozygosity in the genes responsible for sexual pheromones (*MAT*α and *MAT*a), which are situated on chromosome III in both species. Such a loss enables a diploid *MAT*α/*MAT*a strain to transition into either a *MAT*α or *MAT*a diploid state, allowing them to mate with yeasts of the opposite mating type (36, 52). To quantify the degree of hybridization success between *S. cerevisiae* and *S. eubayanus* and its consequential impact on fermentation performance and the profile of volatile compounds in newly formed lager hybrids, we meticulously selected parental for rare mating from candidates spanning a diverse array of lineages from both species (28, 29). Our results indicate that hybridization success between diploid *S. eubayanus* and *S. cerevisiae* is common and particularly high in individual lineages, depending on the species. Although we did not detect differences between *S. eubayanus* lineages, distinct differences were evident between lineages within *S. cerevisiae*. We identified parental strains from the Beer and Bioethanol *S. cerevisiae* lineages, exhibiting the highest hybridization rates. However, it is noteworthy that no hybrids were produced from one Bioethanol lineage strain. Although we did not investigate the underlying reasons for this observation, we believe that certain genetic incompatibilities identified in interspecific hybridizations, particularly those related to mitochondrial and nuclear interactions, might account for the disparities in the mating rates of our hybrids (53–56). In addition, testing a wider range of *S. cerevisiae* strains from different habits might indicate whether industrial or natural environments could impact hybridization rates. Our ploidy analysis revealed that rare mating led to a higher proportion of triploid hybrids. The observed pattern likely arises from two distinct mating scenarios: the mating between spores and diploid parents, which gives rise to triploids, and the mating between haploid spores, resulting in diploids (20, 49, 50). Different studies have contrasting evidence on the effects of yeast ploidy levels on the hybrid's fitness under varying conditions. For instance, Krogerus et al. (22) reported that a tetraploid *de novo* lager hybrid displayed a higher fermentative capacity than a triploid and diploid counterpart. However, this study only used a limited set of hybrids. Interestingly, our study did not reveal a correlation between ploidy level and fermentative capacity across our interspecific hybrids. However, we did observe greater phenotypic variability in triploid hybrids compared with other ploidy levels. This observation may be biased due to the larger number of triploid hybrids in our study, and future research should include a more balanced set of diploid and tetraploid hybrids to assess this relationship more accurately. Our data suggest that ploidy might not be the main factor driving fermentative capacity vigor in *de novo* lager hybrids. However, our phenotypic screening highlighted differences between the hybrid's parental lineages, particularly in *S. cerevisiae*. Interestingly, the observed differences in ethanol tolerance between inter-specific hybrids obtained from the *S. cerevisiae* Beer lineage compared with those obtained from the Wine lineage are likely influenced by the specific environmental conditions to which these strains are naturally adapted (28). Strains belonging to Beer lineages have been historically selected for fermentation at lower ethanol concentrations and may not develop the same level of ethanol tolerance as Wine lineages (15, 57). Wine strains are often exposed to higher ethanol concentrations during grape must fermentation, emphasizing that *S. cerevisiae* lineage-specific phenotypic differences shape the phenotypic traits of interspecific yeast hybrids, such as their capacity to tolerate and thrive in ethanol-rich environments.

Hybridization can generate hybrids with novel traits compared with parental species, which often can meet the requirements of the fermented food industry, such as high fermentative capacity in wort (49). The set of hybrids covered the complete phenotypic space comprising parental species. However, no significant heterosis levels were observed. In this sense, the performance of hybrids emerging from the Beer lineage inherited the maltotriose consumption capacity from the corresponding *S. cerevisiae* parent. Maltotriose is a trisaccharide representing the second most abundant sugar source in the wort, serving as a crucial substrate for yeast metabolism and overall fermentation efficiency (58). Indeed, we observed *MAL* genes to be upregulated in the HB6 strain, which contained a 2 n and 1 n composition of *S. cerevisiae* and *S. eubayanus*

subgenomes, respectively, compared with the HB41 hybrid. In contrast, HB41, carrying an opposite subgenome pattern and a different *S. cerevisiae* parental strain, exhibited upregulation of genes associated with glucose metabolism. Beer ale strains have been historically selected as the preferred yeast to generate hybrid offspring with increased maltotriose consumption (50, 59). Certain hybrid combinations of negative maltotriose strains could increase the metabolization of this trisaccharide, revealing subgenome crosstalk (60). However, our study did not evidence hybrid vigor, possibly attributable to inefficient maltotriose transporters in the parental strains or a lack of crosstalk between the subgenomes for this trait.

The current ambition to develop innovative lager yeast strains seeks to modernize aroma profiles in lager beers. In this context, hybridization emerges as a promising strategy for creating yeasts with distinctive aroma profiles (19, 20, 22, 61). Previous studies have described that hybrids with higher DNA content produce higher concentrations of flavor-active esters (22). However, we generally did not identify a significant correlation between the hybrid's ploidy and VCs best parent heterosis. Overall, these results indicate that ploidy might not significantly affect brewing traits. Instead, when evaluating the contribution of the species' parental lineages, we found that the hybrid's volatile compound profile primarily differs depending on the *S. cerevisae* parental lineage rather than the *S. eubayanus* lineage. Most notably, hybrids emerging from the Beer and Bioethanol lineages differentiated each other in the production of acetate esters and higher alcohols (fruity/flowery), fatty acids (waxy), and their derivative esters (flowery). Although waxy aromas are considered off-flavors in beer, octanoic and decanoic acids in our Bioethanol hybrids were detected within the range of commercial beverages (62, 63). The volatile compound profile of *S. cerevisiae* Beer hybrids varied depending on the *S. eubayanus* parent, suggesting that the volatile compound machinery of *S. eubayanus* exhibits a dominant inheritance over that of the Beer *S. cerevisiae* strain in shaping these traits. Future studies assessing the impact of a larger set of lineages, including domesticated and non-domesticated *S. cerevisiae* lineages, might provide further evidence on the VC trait inheritance pattern. Inherited phenotypes from one parental strain have been documented in hybrid studies (61, 64, 65). In this context, the superior parent heterosis in volatile compound production may stem from the metabolic rewiring in interspecific hybrids (66). The inheritance patterns observed in our study may differ from those found in other species, likely due to the influences of natural selection and human domestication for specific traits (67). In many species of biotechnological interest, heterosis is a common phenomenon in hybrids (67). For example, certain *Brassica napus* allopolyploids between diploid species exhibit higher oil content and improved oil composition than their parent species (68). Conversely, in Arabidopsis, only a few hybrid combinations enhance growth and cold tolerance vigor (69, 70). Although the inheritance patterns in our yeast hybrids reflect the broader principles observed across other organisms, they also offer insights into how specific lineages can contribute to hybrid organisms' overall fitness and industrial utility. Our results indicate that yeast interspecies hybridization is insufficient to achieve fermentative vigor, a trait in lager yeast that may largely be attributed to human domestication.

Our VC pattern analysis identified two hybrid groups with distinct aroma profiles. Hybrids from beer ancestry demonstrated high production of 2-phenyl alcohol, isoamyl alcohol ethyl acetate, and isoamyl acetate. Moreover, heterosis was detected in the production of the last two VCs. The representative *de novo* hybrid HB6 exhibited increased gene expression levels of genes related to alcohol metabolism, such as *ADH2* and *ALD6*, which catalyze acetate synthesis (71). The acetyl-CoA form of acetate is a precursor of acetate esters (72). In this sense, the overexpression of *ALD6* has been used to increase the production of ethyl acetate under fermentation conditions (73). In addition, we found *ILV6* upregulated in HB6, a subunit of the complex acetolactate synthase that increases the activity of *ILV2* in branched-chain amino acid biosynthesis in mitochondria (74). Precursors of this pathway are used to synthesize isoamyl acetate (72). On the contrary, hybrids from Bioethanol ancestry, such as HB41, exhibited a high

production of octanoic acid, decanoic acid, and their respective ethyl esters. These compounds may have resulted from stuck fermentations in beer wort, as observed in the lower fermentative capacity compared with hybrids from beer strains (75). The excretion of these compounds into the media has been associated with detoxification mechanisms (76, 77). Furthermore, previous studies have shown that ethanol enhances the toxicity of octanoic and decanoic acids within the cell (78). In this sense, expression analysis in HB41 evidenced the upregulation of genes involved in β-oxidation such as *POT1* and *ECI1* (79). In this way, hybrid strains derived from the *S. cerevisiae* Bioethanol lineage might exhibit higher fatty acid metabolism as a response to tolerate the alcoholic environment. For instance, HB41 exhibited upregulation in the gene expression of *MGA2* and *IZH4*, both associated with ergosterol biosynthesis, a well-known strategy that yeasts employ to tolerate alcoholic stress (80).

In conclusion, our study emphasizes the importance of genetic diversity in the parental strain background, as demonstrated by the varied hybridization success rates and phenotypic outcomes when different lineages of *S. cerevisiae* and *S. eubayanus* are crossed. Interestingly, we revealed that ploidy levels among the hybrids might not be the primary factor influencing fermentative performance. Instead, we identified a significant relationship between the specific *S. cerevisiae* parental lineage and the hybrids' volatile compound and fermentative profiles, indicating lineage-specific inheritance of traits crucial for the brewing industry. Future research should focus on understanding the genetic and molecular basis of hybrid traits to optimize yeast strains for specific brewing requirements, potentially leading to the development of yeasts with tailor-made characteristics for improved fermentation and aroma profiles.

## ACKNOWLEDGMENTS

We acknowledge Fundación Ciencia & Vida for providing infrastructure, laboratory space, and experiment equipment. This research was partially supported by the supercomputing infrastructure of the National Laboratory for High Performance Computing Chile (NLHPC, ECM-02).

This research was funded by Agencia Nacional de Investigación y Desarrollo (ANID) FONDECYT program and ANID-Programa Iniciativa Científica Milenio—ICN17_022 and NCN2021_050. F.A.C. is supported by FONDECYT grant no. 1220026; V.Z. is supported by ANID grant no. 21201566. C.A.V. is supported by FONDECYT INICIACIÓN grant no. 11230724. A.Q. thanks the Spanish Government, ref. MCIN/AEI/10.13039/501100011033, as IATA (CSIC) "Severo Ochoa" Center of Excellence (CEX2021-001189-S) and MCIU/AEI/FEDER grant references PID2021-126380OB-C31.

## AUTHOR AFFILIATIONS

[1]Universidad de Santiago de Chile, Facultad de Química y Biología, Departamento de Biología, Santiago, Chile

[2]Millennium Institute for Integrative Biology (iBio), Santiago, Chile

[3]Departamento de Biotecnología de los Alimentos, Grupo de Biología de Sistemas en Levaduras de Interés Biotecnológico, Instituto de Agroquímica y Tecnología de los Alimentos (IATA)-CSIC, Valencia, Spain

[4]Millennium Nucleus of Patagonian Limit of Life (LiLi), Valdivia, Chile

[5]Centro de Biotecnología de los Recursos Naturales (CENBio), Facultad de Ciencias Agrarias y Forestales, Universidad Católica del Maule, Talca, Chile

## AUTHOR ORCIDs

Vasni Zavaleta  http://orcid.org/0009-0006-5159-1113
Amparo Querol  http://orcid.org/0000-0002-6478-6845
Francisco A. Cubillos  http://orcid.org/0000-0003-3022-469X

mSystems

## FUNDING

| Funder | Grant(s) | Author(s) |
|---|---|---|
| Agencia Nacional de Investigación y Desarrollo (ANID) | ICN17_022 | Francisco A. Cubillos |
| Agencia Nacional de Investigación y Desarrollo (ANID) | NCN2021_050 | Francisco A. Cubillos |
| ANID \| Fondo Nacional de Desarrollo Científico y Tecnológico (FONDE-CYT) | 1220026 | Francisco A. Cubillos |
| Agencia Nacional de Investigación y Desarrollo (ANID) | 21201566 | Vasni Zavaleta |
| ANID \| Fondo Nacional de Desarrollo Científico y Tecnológico (FONDE-CYT) | 11230724 | Carlos A. Villarroel |
| Ministerio de Ciencia e Innovación (MCIN) | MCIN/AEI/ 10.13039/501100011033 | Amparo Querol |

## AUTHOR CONTRIBUTIONS

Vasni Zavaleta, Formal analysis, Investigation, Methodology, Writing – original draft, Writing – review and editing | Laura Pérez-Través, Investigation, Methodology | Luis A. Saona, Investigation, Methodology | Carlos A. Villarroel, Formal analysis, Investigation, Methodology | Amparo Querol, Formal analysis, Investigation, Methodology, Project administration, Resources | Francisco A. Cubillos, Conceptualization, Formal analysis, Funding acquisition, Investigation, Project administration, Resources, Writing – original draft, Writing – review and editing

## DATA AVAILABILITY

All fastq sequences were deposited in the National Center for Biotechnology Information (NCBI) as a Sequence Read Archive under the BioProject accession number PRJNA1103204.

## ETHICS APPROVAL

This article does not contain any studies with human nor animal subjects performed by any of the authors.

## ADDITIONAL FILES

The following material is available online.

### Supplemental Material

**Supplemental Figures (mSystems00762-24-s0001.pdf).** Figures S1 to S11.
**Supplemental legends (mSystems00762-24-s0002.docx).** Legends to supplemental tables and figures.
**Supplemental Tables (mSystems00762-24-s0003.xlsx).** Tables S1 to S15.

### Open Peer Review

**PEER REVIEW HISTORY (review-history.pdf).** An accounting of the reviewer comments and feedback.

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
