## [Reviewer comments · mSystems]

Understanding brewing trait inheritance in de novo lager yeast hybrids

Francisco Cubillos, Vasni Zavaleta, Laura Pérez-Través, Luis Saona, Carlos Villarroel, and Amparo Querol

Corresponding Author(s): Francisco Cubillos, Universidad de Santiago de Chile

Review Timeline:

Submission Date:	June 3, 2024
Editorial Decision:	August 6, 2024
Revision Received:	September 4, 2024
Accepted:	October 15, 2024

Editor: John Gibbons

Reviewer(s): Disclosure of reviewer identity is with reference to reviewer comments included in decision letter(s). The following individuals involved in review of your submission have agreed to reveal their identity: carla Bautista Rodriguez (Reviewer #1)

Transaction Report:

DOI: <https://doi.org/10.1128/msystems.00762-24>

Re: mSystems00762-24 (Understanding brewing trait inheritance in de novo lager yeast hybrids)

Dear Dr. Francisco A Cubillos:

Thanks so much for your submission to mSystems and for your patience as we secured three reviewers. I am in agreement with the reviewers that this is a comprehensive and interesting manuscript.

I also agree with reviewer 1 that improving the strain nomenclature throughout the manuscript and color usage in figures would greatly enhance clarity. There are also some instances where multiple reviewers thought the writing was confusing and could be improved for clarity. Please make sure to carefully consider all reviewers comments in your resubmission. Looking forward to your revised manuscript.

Revision Guidelines

Sincerely,
John Gibbons
Editor
mSystems

Reviewer #1 (Comments for the Author):

Zavaleta et al. present a compelling study that examines the performance of various de novo lager hybrid crosses between *S. eubayanus* and *S. cerevisiae*, with an industrial focus on fermentation traits. *S. pastorianus*, widely used in lager fermentation, is the result of an ancient hybridization event between these two species. However, the study of hybridization patterns and inheritance has been limited by the lack of diversity in the analyzed *S. eubayanus* parental strains. This paper represents a significant advancement by utilizing a diverse set of *S. eubayanus* strains from different genetic lineages, isolated from native Chilean forests. The primary objective of this work is to uncover the genetic bases of brewing trait inheritance by comparing these de novo hybrids with the parental strains (*S. eubayanus* and *S. cerevisiae*). Several results stand out, making this manuscript an interesting and innovative contribution:

- 1) Firstly, the study observed a differential hybridization success rate among the various crosses, highlighting cross-genetic effects where incompatibilities may play a significant role.
- 2) Interestingly and novelly, when comparing the performance of strains with different ploidy levels-diploids, triploids, and tetraploids-no notable differences were observed.
- 3) Contrary to expectations, although the hybrids encompassed the entire phenotypic range of the parental species, no heterosis was detected.
- 4) Transcriptomics reveals two distinct patterns among hybrids, depending on the dominant parental subgenome, each following different pathways that are analyzed in detail.
- 5) Lineage-specific inheritance was detected for brewing traits, with maltotriose consumption capacity primarily inherited from the corresponding *S. cerevisiae* parent in hybrids derived from the *S. cerevisiae* Beer lineage. However, these same hybrids appear to exhibit dominant inheritance from the *S. eubayanus* parent in terms of volatile compound machinery. This cross-dominance illustrates the intricate genetic interplay shaping fermentation traits. Thus, this study highlights complex heritability patterns in hybrids that contribute to crucial fermentation traits such as aroma.

This study is particularly interesting for industries where hybrids play a crucial role. It also aids in understanding the evolutionary dynamics of fermentation trait development and the influence of inheritance patterns on hybrid performance.

Overall, the manuscript is clearly written, and the experiments and analyses are well-executed and appropriate. The interpretation of the results depicted in the figures is straightforward. While I find the manuscript interesting and the results well-founded, I have significant concerns that I would like to emphasize.

Major

1. The main concern lies in the complexity of the names for all the strains, parentals, hybrids, etc. In my opinion, this makes it difficult to read and interpret the results. Hybrids originating from a parental species are often confused with the parental species themselves. For example, in Figure 2B and 2C, it is unclear whether the x-axis refers to the hybrids. To improve clarity, consider adding a clear axis title like "Hybrids." However, I strongly encourage you to simplify the naming conventions throughout the manuscript by:

- a) Standardizing the nomenclature, such as using Sc_beer, Sc, and Seu for parents, and Hyb_beer for hybrids, would help the reader better understand the results. Consistent nomenclature would make it easier to follow the manuscript without having to frequently refer back to identify the strain and species being discussed. I understand that for the authors, the current nomenclature may seem obvious because it has been internalized over time. However, for the reader, these names hold the same value here and could be changed to a more accessible format to improve clarity and comprehension.
- b) Consistently providing the composition of the hybrid, as you did in Figure 4, using the standardized nomenclature format "Sc_beer (Strain 1 x Strain 2)" throughout the manuscript would enhance clarity. Currently, if this is only applied in some figures, there is potential confusion where other nomenclatures in different figures, like Figure 2, might not clearly indicate hybrids. Adopting this approach uniformly across all figures will ensure that readers can easily identify and understand the hybrids being discussed without ambiguity.

2. Color code across manuscript: With the number of strains, origins, and species covered in the paper, changing colors for each result could easily confuse the reader. For instance, in Figure 3A, I initially thought the yellow-green bar corresponded to the colors used for origins in Figure 2B and D (beer in green, sake in blue, bioethanol in yellow, and wine in red). This misunderstanding led to confusion when I realized in Figure 3A and in the text (L341) that Beer lineage was represented by the yellow bar. Furthermore, this inconsistency continued in Panel 3B where the colors changed again (now beer is red). So I recommend:

- a) To use the same color code when possible.
- b) For heatmaps and clustering, avoid using the same colors as those depicted for lineages (such as beer, etc.). Instead, consider using a gradient of colors with a clear legend positioned next to the heatmap or cluster plot, similar to what is done in Figure 4. This approach ensures clarity and avoids confusion when interpreting the data.

3. Although I find the discussion pertinent and well-structured, the conclusion regarding inheritance patterns seems somewhat superficial. It might be beneficial to expand on this aspect further. Since the analysis of inheritance patterns is a focal point, a more in-depth discussion could be beneficial. For example, while I understand the focus on yeast, exploring inheritance patterns in hybrids of other species used in biotechnology could also be interesting.

4. I found the higher proportion of triploid hybrids really interesting. I wonder if this isn't the cause of observing greater phenotypic variability in them compared to tetraploids (4n). For instance, this variability could be attributed to the higher success rate in triploid formation (29 triploids vs 2 tetraploids), resulting in more phenotypic replicates. Do you think if we had an equivalent number of tetraploid hybrids (4n), we could then discuss comparable phenotypic variability? Perhaps acknowledging the disparity in the number of replicates somewhere would be pertinent.

Minor:

L24: For those unfamiliar with the technique, or if the sentence is isolated from the context of yeast, it might appear to refer to uncommon techniques. Therefore, I suggest adding something like "commonly used in yeast genetics" or providing a bit more context about what this technique involves.

L42: I would suggest changing 'profoundly impacts', which may have negative connotations, to another term, for instance 'increases'.

L75: I wasn't able to find what POF means throughout the manuscript; perhaps clarify that it refers to phenolic off-flavor.

L99: You could be more precise by specifying which field of knowledge it contributes to, for instance, evolutionary dynamics of hybridization inheritance patterns?

L144: Does this mean 6 generations per day? You could clarify by adding the number of generations per day in parentheses along with a reference.

L169: You could add the dilution factor.

L210: In which specific conditions were the differentially expressed genes (DEGs) identified? You mention in L450 beer wort, but what were the exact conditions, such as the percentage of ethanol? It would be helpful to add this detail in the methods section as well.

L253: Fig1 legend: You should add the post-hoc test you used, for example, Tukey. This applies to all post-hoc tests throughout the manuscript; currently, only the ANOVA is mentioned.

L269: Regarding the variation in the effectiveness of the cross combinations: while this is discussed later in the discussion section, a brief explanation in the results could help satisfy the reader's curiosity. For instance, as you pointed out later, the beer clade exhibits the highest success rate in hybridization compared to M9.1, which shows lower effectiveness. This is an interesting observation so it may be appropriate to offer some suggestion or explanation here.

L274: It would be helpful to explain what is meant by 'highest hybridization success' versus 'highest number of hybrid colonies' and why they are not necessarily the same.

L279: Would it be worthwhile to draw a connection here with industrial or natural conditions, where variability in hybrid production can be observed?

L283: Is each point on the graph an average value? If not, the sentence should be corrected.

L327: You conclude with an exception, noting the effect of a triploid on CO₂ production, and then abruptly conclude there is no effect of ploidy overall. Therefore, adding more context, highlighting this interesting exception before returning to the general conclusion, would help improve the flow here.

L343: I'm not sure which bar you are referring to, as it is a boxplot. Are you referring to the green bar in Fig. 3A?

L350 and L353: I understand that in the supplementary figure S5, the range is higher or equal, but in the figure provided in the manuscript (Figure 3D), the hybrids consistently show higher performance. Therefore, this may be confusing for the reader. Additionally, the temperature of 30{degree sign}C does not appear in the main figure. Perhaps you could include something like: "(see Fig S5 for the complete set of strains and temperatures: 4{degree sign}C, 12{degree sign}C, 20{degree sign}C, 25{degree sign}C, 30{degree sign}C, and 37{degree sign}C)" to clarify.

L369: Here, you discuss the first group of hybrids, but I don't see a clear legend in the figure. Although you mention it in parentheses ("yellow bar"), it would be pertinent to include a color legend for clusters in the figure, similar to what is done in Figure 4.

L398: Maybe add a reference.

L400: "We detected similar production levels of 4-VG in hybrids and their corresponding parental strains with no significant differences (p -value > 0.05 , ANOVA, Figure 4B, Table S12), demonstrating a dominant inheritance of this trait." As the sentence is currently written, it is not clear how this would demonstrate dominant heritability.

L414: Add the reference to the figure showing these results.

L415: Here and in other instances, using the verb 'derived' may be confusing, such as in 'hybrids derived from the *S. cerevisiae* NCYC_88 strain'. Instead, you could use 'sharing the parental strain X'.

L416: You concluded that no heterosis was observed, which contrasts with the statement here.

L437: Shouldn't the genetic background be included in your model here as well? There may be combinations of background-ploidy that could confer advantages. In fact, numerous studies have demonstrated genotype-specific effects of ploidy on fitness.

L437: Why is S11 not showing $4n$? If there is a specific reason for excluding them, it should be reported in the text.

L486: I think Figure 6B has some issues:

L458: "The metabolic pathways enriched in each hybrid strain, according to KEGG pathways, indicating HB6's emphasis on 2-oxocarboxylic acid metabolism." The 2-Oxocarboxylic acid metabolism box is white for HB6 (so Gene count 0).

L458: "while HB41's exhibited over-expression on pathways such as lysine biosynthesis and the pentose phosphate pathway, which are integral to cellular building blocks and energy metabolism (Figure 6B)." In this case, the box for the Pentose phosphate pathway for HB41 is also white (so Gene count 0).

L461: You used 'amino acid biosynthetic' in the text, while in the figure it is referred to as 'biosynthesis of amino acids'. Please standardize this to facilitate interpretation of the figure. In the same vein, I can't find anything similar to 'organic acid metabolism' or 'carboxylic biosynthetic' in the figure. Additionally, it is unclear if these categories should be found in panel B because the specific figure you are referring to is not specified.

L461: From line 461 to 469, you stop specifying which figure we should refer to. Since the panel changes, it should be mentioned.

L467: I can't find HXT2 in the figure 6A. Also, I'm unsure if I should find this information in panel A because it's not specified which figure you are referring to.

L479: I can't find MAG2 or IZH4 in the figure 6A. Also, I'm unsure if I should find this information in panel A because it's not specified which figure you are referring to.

Figure 6: L487: Perhaps it would be intuitive to switch the order in the text of the figure caption to HB41 (orange) and HB6 (green), as it appears in this same order in the figure.

L530: I'm not sure I understand the argument of 'However, this study only limited set of hybrids' here, especially because immediately afterwards you provide an example where no differences are detected between the different ploidies either. Perhaps you should expand a bit on what you mean here. It would also be clearer if you could reiterate in terms of ploidy fitness what you found.

Independent comments on figures:

Figure S2B, similar to Figure 2D, appears to show two groups, and the correlation may be influenced by this artifact.

I believe there is an issue with Figure S4. Some points appear to have been swapped between the grids in each graph (in A and B). For example, Bioethanol in A does not have $4n$, but DBVPG6876 in B (the strain in yellow from bioethanol) has a $4n$. The same issue seems to affect Wine; I think Bioethanol and Wine (as well as their strains) are misplaced in some way.

Style:

L70: missing a "." after the references: "consumers attention (Jaeger et al., 2020; Nieto-Villegas et al., 2024)"

L72: missing a space before references: "and fatty acid esters, in beer and other fermented beverages(Krogerus et al., 2016; Mertens et al., 2015; Pérez et al., 2022; Turgeon et al., 2021)".

L130: Reference has different font: "and 30 mg/L lysine (Zaret & Sherman, 1985)."

L164: Reference has different font: "Growthcurver v 0.3.1 (Sprouffske & Wagner, 2016)."

Reviewer #2 (Comments for the Author):

The manuscript by Zavaleta et al., reports the investigation of hybridization success and the impact of hybridization on fermentation performance and volatile compound profiles in newly formed lager hybrids. The authors observed that the hybrids between *S. cerevisiae* and *S. eubayanus* displayed phenotypic variability, notably influenced by maltotriose consumption, and that triploids exhibited greater phenotypic variability. They further showed that while the *S. cerevisiae* parental lineages primarily influenced aroma production, Beer hybrid lineages mostly resembled the corresponding *S. eubayanus* parent. The transcriptome data suggested the dominant inheritance of the *S. eubayanus* aroma profile by the over-expression of genes related to alcohol metabolism and acetate synthesis in hybrids. Based on convincing data generated by systematic analysis, the present study presented interesting information on the complex interactions between parental lineages and hybridization outcomes, which can be usefully applied for creating yeasts with distinct brewing traits through hybridization strategies. There are a few comments to be addressed with additional information and validation data, as followings, to improve the manuscript in more complete way.

Major comment

It is recommended to validate transcriptome analysis data of the polyploid hybrids, HB6 and HB41, by qRT-PCR analysis of a set of DEGs with the primers specific to each parental gene. The lineage specific gene analysis, which can distinguish the genes from *S. eubayanus* or from *S. cerevisiae*, would provide more advanced information on the origin of the up-regulated genes and on the different expression patterns before and after hybridization.

Minor comments

- 1). Please provide information on the source of standard volatile compounds used for quantitation in Material and Methods.
- 2) Please check the list of references to unify the format of presentations.

Reviewer #3 (Comments for the Author):

The paper entitled "Understanding brewing trait inheritance in de novo lager yeast hybrids" by Zavaleta et al. explores the fermentative capacities and characteristics of hybrids between *S. cerevisiae* and *S. eubayanus*. The findings are significant not only from a fundamental perspective on hybridization but also in the applied context of beer product diversification.

While the study is interesting, the manuscript requires revision to improve clarity, objectivity, and engagement, particularly in the sections on volatile compounds, RNA-Seq, and the discussion. Specific comments are outlined below:

Referring to "Beer" lineages can be misleading, as it may be interpreted as referring to the Beer 1 lineage. Since this is not the case, the authors should clarify by using "mixed beer" (at least upon first mention) or another term that avoids confusion while maintaining consistency with the population names used in the original papers.

Line 43: You want to say phenotypes instead of "genotypes"?

Line 45: Maybe replace "...lead to replace parental species..." by 'lead to parental replacement...'

Line 47: "anthropogenic environmental changes" or anthropogenic transitions?

Line 57-58: Needs revision.

Line 63: You need to refer which hybrids. *S. pastorianus* hybrids combine...

Line 145: Why 42 generations? Has this been proven to be enough for genome stabilization?

Line 162-164: Why use different wavelengths for adjusting the inoculum and to take the measures during the experiment?

Line 164 - 165: When you mention the average, do you mean the average from the replicates for each strain? Needs clarification.

Line 200: In which conditions were the cells growth and at which stage were they harvested for RNA extraction? I think this information should be added here.

PCA analysis despite being presented in a main figure is not mentioned in the methods. Please add.

Line 235: From Table S2 it seems to be maltotriose consumption. Not 100% as for maltose, but some are above 15%. This sentence needs to be reformulated.

Line 250: "...from each parent..." doesn't make sense in this sentence.

Line 261: Instead of "different letters" you mean 'different numbers'? And the numbers don't reflect the number of hybrids obtained? This part of the legend needs to be clarified.

Line 280: I think "most *S. cerevisiae* lineages" is too strong, because only a handful of lineages were tested.

Line 331: Can this difference be explained by different parental species rather than ploidy?

Line 336: In this section are mentioned the differences between hybrids derived from the beer lineage and those from the wine lineage, with the latter performing better in 12% ethanol. This result should be further explored, as it is closely related to the environmental conditions to which these strains are naturally adapted. A discussion on how these environmental adaptations influence the observed ethanol tolerance should be added.

526: Shouldn't be 'strains' instead of "species"?

Line 539: Peter et al. 2018 is not the best example to compare here, as this study was not done with hybrids.

Line 554: It is worth mentioning here that the *S. cerevisiae* parental strain is also different between these two hybrids, which certainly also may affect.

Line 556: This sentence doesn't relate to your results and may generate confusion as the hybrids you discuss here (HB6 and HB41) don't have wine subgenomes.

Line 567-569: This part needs to be rewritten. Is confusing and repetitive.

Line 608-609: I think is worth mentioning and reinforcing that is mostly influenced by *S. cerevisiae* parental strain.

Review

Zavaleta et al. present a compelling study that examines the performance of various de novo lager hybrid crosses between *S. eubayanus* and *S. cerevisiae*, with an industrial focus on fermentation traits. *S. pastorianus*, widely used in lager fermentation, is the result of an ancient hybridization event between these two species. However, the study of hybridization patterns and inheritance has been limited by the lack of diversity in the analyzed *S. eubayanus* parental strains. This paper represents a significant advancement by utilizing a diverse set of *S. eubayanus* strains from different genetic lineages, isolated from native Chilean forests. The primary objective of this work is to uncover the genetic bases of brewing trait inheritance by comparing these de novo hybrids with the parental strains (*S. eubayanus* and *S. cerevisiae*). Several results stand out, making this manuscript an interesting and innovative contribution:

- 1) Firstly, the study observed a differential hybridization success rate among the various crosses, highlighting cross-genetic effects where incompatibilities may play a significant role.
- 2) Interestingly and novelly, when comparing the performance of strains with different ploidy levels—diploids, triploids, and tetraploids—no notable differences were observed.
- 3) Contrary to expectations, although the hybrids encompassed the entire phenotypic range of the parental species, no heterosis was detected.
- 4) Transcriptomics reveals two distinct patterns among hybrids, depending on the dominant parental subgenome, each following different pathways that are analyzed in detail.
- 5) Lineage-specific inheritance was detected for brewing traits, with maltotriose consumption capacity primarily inherited from the corresponding *S. cerevisiae* parent in hybrids derived from the *S. cerevisiae* Beer lineage. However, these same hybrids appear to exhibit dominant inheritance from the *S. eubayanus* parent in terms of volatile compound machinery. This cross-dominance illustrates the intricate genetic interplay shaping fermentation traits. Thus, this study highlights complex heritability patterns in hybrids that contribute to crucial fermentation traits such as aroma.

This study is particularly interesting for industries where hybrids play a crucial role. It also aids in understanding the evolutionary dynamics of fermentation trait development and the influence of inheritance patterns on hybrid performance.

Overall, the manuscript is clearly written, and the experiments and analyses are well-executed and appropriate. The interpretation of the results depicted in the figures is straightforward. While I find the manuscript interesting and the results well-founded, I have significant concerns that I would like to emphasize.

Major

1. The main concern lies in the **complexity of the names** for all the strains, parentals, hybrids, etc. In my opinion, this makes it difficult to read and interpret the results. Hybrids originating from a parental species are often confused with the parental species themselves. For example, in Figure 2B and 2C, it is unclear whether the x-axis refers to the hybrids. To improve clarity, consider adding a clear axis title like "Hybrids." However, I strongly encourage you to simplify the naming conventions throughout the manuscript by:
 - a. Standardizing the nomenclature, such as using Sc_beer, Sc, and Seu for parents, and Hyb_beer for hybrids, would help the reader better understand the results. Consistent nomenclature would make it easier to follow the manuscript without having to frequently refer back to identify the strain and species being discussed. I understand that for the authors, the current nomenclature may seem obvious because it has been internalized over time. However, for the reader, these names hold the same value here and could be changed to a more accessible format to improve clarity and comprehension.
 - b. Consistently providing the composition of the hybrid, as you did in Figure 4, using the standardized nomenclature format "Sc_beer (Strain 1 x Strain 2)" throughout the manuscript would enhance clarity. Currently, if this is only applied in some figures, there is potential confusion where other nomenclatures in different figures, like Figure 2, might not clearly indicate hybrids. Adopting this approach uniformly across all figures will ensure that readers can easily identify and understand the hybrids being discussed without ambiguity.

2. **Color code across manuscript:** With the number of strains, origins, and species covered in the paper, changing colors for each result could easily confuse the reader. For instance, in Figure 3A, I initially thought the yellow-green bar corresponded to the colors used for origins in Figure 2B and D (beer in green, sake in blue, bioethanol in yellow, and wine in red). This misunderstanding led to confusion when I realized in Figure 3A and in the text (L341) that Beer lineage was represented by the yellow bar. Furthermore, this inconsistency continued in Panel 3B where the colors changed again (now beer is red). So I recommend:
 - a. To use the same color code when possible.
 - b. For heatmaps and clustering, avoid using the same colors as those depicted for lineages (such as beer, etc.). Instead, consider using a gradient of colors with a clear legend positioned next to the heatmap or cluster plot, similar to what is done in Figure 4. This approach ensures clarity and avoids confusion when interpreting the data.

3. Although I find the discussion pertinent and well-structured, the conclusion regarding inheritance patterns seems somewhat superficial. It might be beneficial to expand on this aspect further. Since the analysis of inheritance patterns is a focal point, a more in-depth discussion could be beneficial. For example, while I understand the focus on yeast, exploring inheritance patterns in hybrids of other species used in biotechnology could also be interesting.
4. I found the higher proportion of triploid hybrids really interesting. I wonder if this isn't the cause of observing greater phenotypic variability in them compared to tetraploids (4n). For instance, this variability could be attributed to the higher success rate in triploid formation (29 triploids vs 2 tetraploids), resulting in more phenotypic replicates. Do you think if we had an equivalent number of tetraploid hybrids (4n), we could then discuss comparable phenotypic variability? Perhaps acknowledging the disparity in the number of replicates somewhere would be pertinent.

Minor:

5. **L24:** For those unfamiliar with the technique, or if the sentence is isolated from the context of yeast, it might appear to refer to uncommon techniques. Therefore, I suggest adding something like "commonly used in yeast genetics" or providing a bit more context about what this technique involves.
6. **L42:** I would suggest changing 'profoundly impacts', which may have negative connotations, to another term, for instance 'increases'.
7. **L75:** I wasn't able to find what POF means throughout the manuscript; perhaps clarify that it refers to phenolic off-flavor.
8. **L99:** You could be more precise by specifying which field of knowledge it contributes to, for instance, evolutionary dynamics of hybridization inheritance patterns?
9. **L144:** Does this mean 6 generations per day? You could clarify by adding the number of generations per day in parentheses along with a reference.
10. **L169:** You could add the dilution factor.
11. **L210:** In which specific conditions were the differentially expressed genes (DEGs) identified? You mention in L450 beer wort, but what were the exact conditions, such as the percentage of ethanol? It would be helpful to add this detail in the methods section as well.
12. **L253:** Fig1 legend: You should add the post-hoc test you used, for example, Tukey. This applies to all post-hoc tests throughout the manuscript; currently, only the ANOVA is mentioned.

13. **L269:** Regarding the variation in the effectiveness of the cross combinations: while this is discussed later in the discussion section, a brief explanation in the results could help satisfy the reader's curiosity. For instance, as you pointed out later, the beer clade exhibits the highest success rate in hybridization compared to M9.1, which shows lower effectiveness. This is an interesting observation so it may be appropriate to offer some suggestion or explanation here.
14. **L274:** It would be helpful to explain what is meant by 'highest hybridization success' versus 'highest number of hybrid colonies' and why they are not necessarily the same.
15. **L279:** Would it be worthwhile to draw a connection here with industrial or natural conditions, where variability in hybrid production can be observed?
16. **L283:** Is each point on the graph an average value? If not, the sentence should be corrected.
17. **L327:** You conclude with an exception, noting the effect of a triploid on CO₂ production, and then abruptly conclude there is no effect of ploidy overall. Therefore, adding more context, highlighting this interesting exception before returning to the general conclusion, would help improve the flow here.
18. **L343:** I'm not sure which bar you are referring to, as it is a boxplot. Are you referring to the green bar in Fig. 3A?
19. **L350 and L353:** I understand that in the supplementary figure S5, the range is higher or equal, but in the figure provided in the manuscript (Figure 3D), the hybrids consistently show higher performance. Therefore, this may be confusing for the reader. Additionally, the temperature of 30°C does not appear in the main figure. Perhaps you could include something like: "(see Fig S5 for the complete set of strains and temperatures: 4°C, 12°C, 20°C, 25°C, 30°C, and 37°C)" to clarify.
20. **L369:** Here, you discuss the first group of hybrids, but I don't see a clear legend in the figure. Although you mention it in parentheses ("yellow bar"), it would be pertinent to include a color legend for clusters in the figure, similar to what is done in Figure 4.
21. **L398:** Maybe add a reference.
22. **L400:** *"We detected similar production levels of 4-VG in hybrids and their corresponding parental strains with no significant differences (p-value > 0.05, ANOVA, **Figure 4B, Table S12**), demonstrating a dominant inheritance of this trait."* As the sentence is currently written, it is not clear how this would demonstrate dominant heritability.
23. **L414:** Add the reference to the figure showing these results.
24. **L415:** Here and in other instances, using the verb 'derived' may be confusing, such as in '*hybrids derived from the S. cerevisiae NCYC_88 strain*'. Instead, you could use 'sharing the parental strain X'.

25. **L416:** You concluded that no heterosis was observed, which contrasts with the statement here.
26. **L437:** Shouldn't the genetic background be included in your model here as well? There may be combinations of background-ploidy that could confer advantages. In fact, numerous studies have demonstrated genotype-specific effects of ploidy on fitness.
27. **L437:** Why is S11 not showing 4n? If there is a specific reason for excluding them, it should be reported in the text.
28. **L486:** I think Figure 6B has some issues:
- L458:** "*The metabolic pathways enriched in each hybrid strain, according to KEGG pathways, indicating HB6's emphasis on 2-oxocarboxylic acid metabolism.*" The 2-Oxocarboxylic acid metabolism box is white for HB6 (so Gene count 0).
 - L458:** "*while HB41's exhibited over-expression on pathways such as lysine biosynthesis and the pentose phosphate pathway, which are integral to cellular building blocks and energy metabolism (Figure 6B).*" In this case, the box for the *Pentose phosphate pathway* for HB41 is also white (so Gene count 0).
 - L461:** You used 'amino acid biosynthetic' in the text, while in the figure it is referred to as 'biosynthesis of amino acids'. Please standardize this to facilitate interpretation of the figure. In the same vein, I can't find anything similar to 'organic acid metabolism' or 'carboxylic biosynthetic' in the figure. Additionally, it is unclear if these categories should be found in panel B because the specific figure you are referring to is not specified.
29. **L461:** From line 461 to 469, you stop specifying which figure we should refer to. Since the panel changes, it should be mentioned.
30. **L467:** I can't find *HXT2* in the figure 6A. Also, I'm unsure if I should find this information in panel A because it's not specified which figure you are referring to.
31. **L479:** I can't find *MAG2* or *IZH4* in the figure 6A. Also, I'm unsure if I should find this information in panel A because it's not specified which figure you are referring to.
32. **Figure 6: L487:** Perhaps it would be intuitive to switch the order in the text of the figure caption to HB41 (orange) and HB6 (green), as it appears in this same order in the figure.
33. **L530:** I'm not sure I understand the argument of '*However, this study only limited set of hybrids*' here, especially because immediately afterwards you provide an example where no differences are detected between the different ploidy levels either. Perhaps you should expand a bit on what you mean here. It would also be clearer if you could reiterate in terms of ploidy fitness what you found.

Independent comments on figures:

34. Figure S2B, similar to Figure 2D, appears to show two groups, and the correlation may be influenced by this artifact.
35. I believe there is an issue with **Figure S4**. Some points appear to have been swapped between the grids in each graph (in A and B). For example, Bioethanol in A does not have 4n, but DBVPG6876 in B (the strain in yellow from bioethanol) has a 4n. The same issue

seems to affect Wine; I think Bioethanol and Wine (as well as their strains) are misplaced in some way.

Style:

36. **L70:** missing a "." after the references: "consumers attention (Jaeger et al., 2020; Nieto-Villegas et al., 2024)"
37. **L72:** missing a space before references: "and fatty acid esters, in beer and other fermented beverages(Krogerus et al., 2016; Mertens et al., 2015; Pérez et al., 2022; Turgeon et al., 2021)".
38. **L130:** Reference has different font: "and 30 mg/L lysine (Zaret & Sherman, 1985)."
39. **L164:** Reference has different font: "Growthcurver v 0.3.1 (Sprouffske & Wagner, 2016)."

Reviewer #1 (Comments for the Author):

Zavaleta et al. present a compelling study that examines the performance of various de novo lager hybrid crosses between *S. eubayanus* and *S. cerevisiae*, with an industrial focus on fermentation traits. *S. pastorianus*, widely used in lager fermentation, is the result of an ancient hybridization event between these two species. However, the study of hybridization patterns and inheritance has been limited by the lack of diversity in the analyzed *S. eubayanus* parental strains. This paper represents a significant advancement by utilizing a diverse set of *S. eubayanus* strains from different genetic lineages, isolated from native Chilean forests. The primary objective of this work is to uncover the genetic bases of brewing trait inheritance by comparing these de novo hybrids with the parental strains (*S. eubayanus* and *S. cerevisiae*). Several results stand out, making this manuscript an interesting and innovative contribution:

- 1) Firstly, the study observed a differential hybridization success rate among the various crosses, highlighting cross-genetic effects where incompatibilities may play a significant role.
- 2) Interestingly and novelly, when comparing the performance of strains with different ploidy levels-diploids, triploids, and tetraploids-no notable differences were observed.
- 3) Contrary to expectations, although the hybrids encompassed the entire phenotypic range of the parental species, no heterosis was detected.
- 4) Transcriptomics reveals two distinct patterns among hybrids, depending on the dominant parental subgenome, each following different pathways that are analyzed in detail.
- 5) Lineage-specific inheritance was detected for brewing traits, with maltotriose consumption capacity primarily inherited from the corresponding *S. cerevisiae* parent in hybrids derived from the *S. cerevisiae* Beer lineage. However, these same hybrids appear to exhibit dominant inheritance from the *S. eubayanus* parent in terms of volatile compound machinery. This cross-dominance illustrates the intricate genetic interplay shaping fermentation traits. Thus, this study highlights complex heritability patterns in hybrids that contribute to crucial fermentation traits such as aroma.

This study is particularly interesting for industries where hybrids play a crucial role. It also aids in understanding the evolutionary dynamics of fermentation trait development and the influence of inheritance patterns on hybrid performance.

Overall, the manuscript is clearly written, and the experiments and analyses are well-executed and appropriate. The interpretation of the results depicted in the figures is straightforward. While I find the manuscript interesting and the results well-founded, I have significant concerns that I would like to emphasize.

R: First of all, we would like to thank the reviewer for the helpful and constructive comments on our manuscript. We have now addressed all comments and made the suggested changes.

Major

1. The main concern lies in the complexity of the names for all the strains, parentals, hybrids, etc. In my opinion, this makes it difficult to read and interpret the results. Hybrids originating from a parental species are often confused with the parental species themselves. For example, in Figure 2B and 2C, it is unclear whether the x-axis refers to the hybrids. To improve clarity, consider adding a clear axis title like "Hybrids." However, I strongly encourage you to simplify the naming conventions throughout the manuscript by:

a) Standardizing the nomenclature, such as using Sc_beer, Sc, and Seu for parents, and Hyb_beer for hybrids, would help the reader better understand the results. Consistent nomenclature would make it easier to follow the manuscript without having to frequently refer back to identify the strain and species being discussed. I understand that for the authors, the current nomenclature may seem obvious because it has been internalized over time. However, for the reader, these names hold the same value here and could be changed to a more accessible format to improve clarity and comprehension.

R: We have now addressed this and provided a standardized nomenclature for species and hybrids. We followed the guidelines suggested by the reviewer, and the nomenclature is available in the Supplementary Tables (see S1 and Figure 3 for examples).

b) Consistently providing the composition of the hybrid, as you did in Figure 4, using the standardized nomenclature format "Sc_beer (Strain 1 x Strain 2)" throughout the manuscript would enhance clarity. Currently, if this is only applied in some figures, there is potential confusion where other nomenclatures in different figures, like Figure 2, might not clearly indicate hybrids. Adopting this approach uniformly across all figures will ensure that readers can easily identify and understand the hybrids being discussed without ambiguity.

R: This has been done throughout the manuscript

2. Color code across manuscript: With the number of strains, origins, and species covered in the paper, changing colors for each result could easily confuse the reader. For instance, in Figure 3A, I initially thought the yellow-green bar corresponded to the colors used for origins in Figure 2B and D (beer in green, sake in blue, bioethanol in yellow, and wine in red). This misunderstanding led to confusion when I realized in Figure 3A and in the text (L341) that Beer lineage was represented by the yellow bar. Furthermore, this inconsistency continued in Panel 3B where the colors changed again (now beer is red). So I recommend:

a) To use the same color code when possible.

b) For heatmaps and clustering, avoid using the same colors as those depicted for lineages (such as beer, etc.). Instead, consider using a gradient of colors with a clear legend positioned next to the heatmap or cluster plot, similar to what is done in Figure 4. This approach ensures clarity and avoids confusion when interpreting the data.

R: We thank the reviewer for the constructive comments on our plots. We have now standardized the colour coding throughout the figures and followed the reviewer's suggestions.

3. Although I find the discussion pertinent and well-structured, the conclusion regarding inheritance patterns seems somewhat superficial. It might be beneficial to expand on this aspect further. Since the analysis of inheritance patterns is a focal point, a more in-depth discussion could be beneficial. For example, while I understand the focus on yeast, exploring inheritance patterns in hybrids of other species used in biotechnology could also be interesting.

R: We have now incorporated an additional paragraph in the discussion section with a more in-depth discussion on inheritance patterns in yeast and other relevant species. These are found in lines 605 -615 and read as follows:

'The inheritance patterns observed in our study may differ from those found in other species, likely due to the influences of natural selection and human domestication for specific traits (Chen et al., 2010). In many species of biotechnological interest, heterosis is a common phenomenon in hybrids (Chen et al., 2010). For example, certain Brassica napus allopolyploids between diploid species exhibit higher oil content and improved oil composition than their parent species (Gu et al., 2024). Conversely, in Arabidopsis, only few hybrid combinations result in enhanced growth and cold tolerance vigour (Meyer et al., 2004; Rhoder et al., 2004). While the inheritance

patterns in our yeast hybrids reflect the broader principles observed across other organisms, they also offer insights into how specific lineages can contribute to hybrid organisms' overall fitness and industrial utility. Our results indicate that yeast interspecies hybridization is insufficient to achieve fermentative vigour, a trait in lager yeast that may largely be attributed to human domestication.

4. I found the higher proportion of triploid hybrids really interesting. I wonder if this isn't the cause of observing greater phenotypic variability in them compared to tetraploids (4n). For instance, this variability could be attributed to the higher success rate in triploid formation (29 triploids vs 2 tetraploids), resulting in more phenotypic replicates. Do you think if we had an equivalent number of tetraploid hybrids (4n), we could then discuss comparable phenotypic variability? Perhaps acknowledging the disparity in the number of replicates somewhere would be pertinent.

R: We agree with the reviewer, and we have now incorporated a sentence commenting on this in the discussion section (lines 551 – 555):

'Interestingly, our study did not reveal a correlation between ploidy level and fermentative capacity across our interspecific hybrids. However, we did observe greater phenotypic variability in triploid hybrids compared to other ploidy levels. This observation may be biased due to the larger number of triploid hybrids in our study, and future research should include a more balanced set of diploid and tetraploid hybrids to assess this relationship more accurately.'

Minor:

L24: For those unfamiliar with the technique, or if the sentence is isolated from the context of yeast, it might appear to refer to uncommon techniques. Therefore, I suggest adding something like "commonly used in yeast genetics" or providing a bit more context about what this technique involves.

R: We have now modified this sentence and reads as follows:

'Polyploid hybrids were generated through a spontaneous diploid hybridization technique (rare-mating), revealing a prevalence of triploids and diploids over tetraploids.'

L42: I would suggest changing 'profoundly impacts', which may have negative connotations, to another term, for instance 'increases'.

R: Changed

L75: I wasn't able to find what POF means throughout the manuscript; perhaps clarify that it refers to phenolic off-flavor.

R: This has been incorporated

L99: You could be more precise by specifying which field of knowledge it contributes to, for instance, evolutionary dynamics of hybridization inheritance patterns?

R: We have changed this sentence accordingly, and now reads:

'This research contributes to our knowledge of hybridization inheritance patterns in lager yeast and explores new avenues for enhancing yeast strains in the biotechnology industry.'

L144: Does this mean 6 generations per day? You could clarify by adding the number of generations per day in parentheses along with a reference.

R: The number of generations is based on the initial and final optical densities (OD), independently of the number of hours or days. We have now incorporated another sentence in parenthesis to clarify this statement. This now reads:

'This cell transfer process was repeated seven times, corresponding to approximately 42 generations (approximately six generations per each transfer passage).'

L169: You could add the dilution factor.

R: Incorporated

L210: In which specific conditions were the differentially expressed genes (DEGs) identified? You mention in L450 beer wort, but what were the exact conditions, such as the percentage of ethanol? It would be helpful to add this detail in the methods section as well.

R: We have incorporated a sentence indicating the specific environmental conditions for RNA-extraction, including the wort composition. Usually, RNA-seq experiments under fermentation conditions are carried out at the initial fermentation stages, when cells adapt to the stressful environment. Furthermore, we cannot depict specific environmental conditions, since this is a fermentation environment where we can only provide the initial conditions. Still, we now provide further details in terms on the experimental conditions for RNA-seq. This sentence now reads:

'RNA was obtained and processed after 24 h under 12°B beer wort fermentation (as indicated in the 'Wort fermentations' subsection) at 12°C in triplicates as previously described (Molinet et al, 2024).'

L253: Fig1 legend: You should add the post-hoc test you used, for example, Tukey. This applies to all post-hoc tests throughout the manuscript; currently, only the ANOVA is mentioned.

R: We have now incorporated this information in the 'Statistical Analysis' section, and every passage throughout the manuscript where needed.

L269: Regarding the variation in the effectiveness of the cross combinations: while this is discussed later in the discussion section, a brief explanation in the results could help satisfy the reader's curiosity. For instance, as you pointed out later, the beer clade exhibits the highest success rate in hybridization compared to M9.1, which shows lower effectiveness. This is an interesting observation so it may be appropriate to offer some suggestion or explanation here.

R: We appreciate the reviewer's suggestion, and we have now incorporated a brief explanation sentence (lines 277-279)

'Among the 36 potential cross combinations, 21 produced at least one positive hybrid colony (Figure 1D). The remaining combinations did not evidence hybrid colonies, likely due to genetic incompatibilities related to mitochondrial and nuclear interactions'

L274: It would be helpful to explain what is meant by 'highest hybridization success' versus 'highest number of hybrid colonies' and why they are not necessarily the same.

R: Hybridization success rate refers to the number of positive attempts relative to the total number of cross-attempts performed. That being said, we found different numbers of hybrid colonies per cross attempt, independently of the success rate. Therefore, in our study classify the strains based on two indexes: 'Hybridization success rate' and 'highest number of hybrid colonies'. We have now incorporated an additional sentence to make this easier to understand.

*'The beer clade *S. cerevisiae* strain NCYC_88 demonstrated the highest hybridization success rate (defined as 'number of positive attempts/ numbers of cross-attempts') among *S. cerevisiae* strains at 52.6% (Table S3). In contrast, the *S. cerevisiae* bioethanol M9.1 strain did not produce hybrids with any *S. eubayanus* parental strain. Among the *S. eubayanus* strains, CL248.1 (PB-2 lineage) had the highest hybridization success rate at 38.9% (Table S3), while CL620.1 represented the strain with the highest number of hybrid colonies (31 hybrids after 5 positive cross-attempts, Figure 1D).'*

L279: Would it be worthwhile to draw a connection here with industrial or natural conditions, where variability in hybrid production can be observed?

*R: It would be nice to connect both since this is an interesting question in the field. However, most of our *S. cerevisiae* strains were obtained from industrial settings, and we cannot address this question properly. Instead, we have incorporated a sentence in the discussion section mentioning this interesting perspective (lines 538 – 544).*

*'While we did not investigate the underlying reasons for this observation, we believe that certain genetic incompatibilities identified in interspecific hybridizations, particularly those related to mitochondrial and nuclear interactions, might account for the disparities in the mating rates of our hybrids (Blanckaert & Payseur, 2021; Lee et al., 2008; Moran et al., 2024; Swamy et al., 2022). In addition, testing a wider range of *S. cerevisiae* strains from different habits might indicate whether industrial or natural environments could impact hybridization rates.'*

L283: Is each point on the graph an average value? If not, the sentence should be corrected.

R: Each dot corresponds to the strain's average value. This is now indicated in the figure legend.

L327: You conclude with an exception, noting the effect of a triploid on CO₂ production, and then abruptly conclude there is no effect of ploidy overall. Therefore, adding more context, highlighting this interesting exception before returning to the general conclusion, would help improve the flow here.

R: We have now changed this sentence, and now reads:

*'Similarly, when examining the influence of *S. cerevisiae* and *S. eubayanus* lineages per ploidy level on fermentative capacity, we found no significant differences (p -value > 0.05, Mann-Whitney-Wilcoxon test, Figures S4). In the analysis at the individual parental strain level, we found that the *S. eubayanus* CL248.1 strain exhibited a notable exception, showing higher CO₂ production in 3n hybrids than 2n hybrids (p -value < 0.05, Mann-Whitney-Wilcoxon test, Figure S4D). However, this corresponded to a particular case not observed in other genetic backgrounds. Altogether, the species and lineage level analysis suggest that the *S. cerevisiae* lineage is the primary determinant of the hybrid's fermentative capacity and that ploidy levels might not influence this trait in the lager's hybrid background.'*

L343: I'm not sure which bar you are referring to, as it is a boxplot. Are you referring to the green bar in Fig. 3A?

R: This sentence refers to Figure 3B. We have removed the mistake (green bar) accordingly.

L350 and L353: I understand that in the supplementary figure S5, the range is higher or equal, but in the figure provided in the manuscript (Figure 3D), the hybrids consistently show higher performance. Therefore, this may be confusing for the reader. Additionally, the temperature of 30°C does not appear in the main figure. Perhaps you could include something like: "(see Fig S5

for the complete set of strains and temperatures: 4°C, 12°C, 20°C, 25°C, 30°C, and 37°C)" to clarify.

R: We have incorporated the suggested sentence in the manuscript. Furthermore, we have incorporated another example in Figure 3D, showing a hybrid with similar phenotypic performance compared to one of the parental strains.

L369: Here, you discuss the first group of hybrids, but I don't see a clear legend in the figure. Although you mention it in parentheses ("yellow bar"), it would be pertinent to include a color legend for clusters in the figure, similar to what is done in Figure 4.

R: We have removed the colour bars from Figure 4. Instead, the Hybrids' parents are now shown in the figure.

L398: Maybe add a reference.

R: Incorporated

L400: "We detected similar production levels of 4-VG in hybrids and their corresponding parental strains with no significant differences (p-value > 0.05, ANOVA, Figure 4B, Table S12), demonstrating a dominant inheritance of this trait." As the sentence is currently written, it is not clear how this would demonstrate dominant heritability.

R: We have deleted the 'dominant inheritance' sentence.

L414: Add the reference to the figure showing these results.

R: Incorporated

L415: Here and in other instances, using the verb 'derived' may be confusing, such as in 'hybrids derived from the *S. cerevisiae* NCYC_88 strain'. Instead, you could use 'sharing the parental strain X'.

R: This word was corrected throughout the manuscript.

L416: You concluded that no heterosis was observed, which contrasts with the statement here.

R: We have now specified that this is for volatile compounds. The sentence now reads:

*'Our analysis revealed 20 hybrids exhibiting BPH for at least one VC, except for the HB28 hybrid. Interestingly, hybrids sharing the *S. cerevisiae* parental strain NCYC_88 strain exhibited 6 out of 14 VCs with BPH, demonstrating the high levels of heterosis for VC production in the novel hybrids.'*

L437: Shouldn't the genetic background be included in your model here as well? There may be combinations of background-ploidy that could confer advantages. In fact, numerous studies have demonstrated genotype-specific effects of ploidy on fitness.

R: Indeed, we agree with the reviewer that this is an interesting observation to pursue. However, we don't have enough power to assess the background-ploidy effect. In the future, assessing a more significant number of hybrids within this set of backgrounds could allow addressing the reviewer's suggestion.

L437: Why is S11 not showing 4n? If there is a specific reason for excluding them, it should be reported in the text.

R: Since only two tetraploid hybrids were available, we could not perform the statistical analysis for Figure S11: This has been included in the text. This sentence now reads (lines 449 – 455):

'Finally, exploring the correlation between ploidy and BPH did not detect an overall significant correlation (p -value > 0.05, Mann-Whitney-Wilcoxon test, Figure S10 and S11). However, some triploid strains exhibited the highest BPH values and greater variance (CV = 0.580) compared to diploids (CV = 0.295) and tetraploids (CV = 0.186). Since only two tetraploid strains were available, we could not compare the volatile compound production on these hybrids to other ploidy levels. These results suggest an increased BPH phenotypic variability among triploid hybrids compared to diploids.'

L486: I think Figure 6B has some issues:

L458: "The metabolic pathways enriched in each hybrid strain, according to KEGG pathways, indicating HB6's emphasis on 2-oxocarboxylic acid metabolism." The 2-Oxocarboxylic acid metabolism box is white for HB6 (so Gene count 0).

R: Corrected.

L458: "while HB41's exhibited over-expression on pathways such as lysine biosynthesis and the pentose phosphate pathway, which are integral to cellular building blocks and energy metabolism (Figure 6B)." In this case, the box for the Pentose phosphate pathway for HB41 is also white (so Gene count 0).

R: Corrected.

L461: You used 'amino acid biosynthetic' in the text, while in the figure it is referred to as 'biosynthesis of amino acids'. Please standardize this to facilitate interpretation of the figure. In the same vein, I can't find anything similar to 'organic acid metabolism' or 'carboxylic biosynthetic' in the figure. Additionally, it is unclear if these categories should be found in panel B because the specific figure you are referring to is not specified.

R: We have revised these pathways and corrected them accordingly.

L461: From line 461 to 469, you stop specifying which figure we should refer to. Since the panel changes, it should be mentioned.

R: We have incorporated the figures we are referring to.

L467: I can't find HXT2 in the figure 6A. Also, I'm unsure if I should find this information in panel A because it's not specified which figure you are referring to.

R: We have incorporated HXT2 and all the genes mentioned in this paragraph in Figure 6A.

L479: I can't find MAG2 or IZH4 in the figure 6A. Also, I'm unsure if I should find this information in panel A because it's not specified which figure you are referring to.

R: We have included those genes in Figure 6A. In addition, we now provide a new Figure 6C showing the impact of various differentially expressed genes on volatile compound production.

Figure 6: L487: Perhaps it would be intuitive to switch the order in the text of the figure caption to HB41 (orange) and HB6 (green), as it appears in this same order in the figure.

R: Changed

L530: I'm not sure I understand the argument of 'However, this study only limited set of hybrids' here, especially because immediately afterwards you provide an example where no differences are detected between the different ploidies either. Perhaps you should expand a bit on what you mean here. It would also be clearer if you could reiterate in terms of ploidy fitness what you found.

R: We have now changed this paragraph to improve the main message out of it. This sentence now reads (lines 547-552):

'Different studies have contrasting evidence on the effects of yeast ploidy levels on the hybrid's fitness under varying conditions. For instance, Krogerus et al. (2016) reported that a tetraploid de novo lager hybrid displayed a higher fermentative capacity than a triploid and diploid counterpart. However, this study only used a limited set of hybrids. In contrast, Peter et al. (2018) found that S. cerevisiae diploids exhibited superior fitness compared to triploids and tetraploids strains. Interestingly, in our study we did not observe a correlation between ploidy and fermentative capacity across our interspecific hybrids. This suggests that ploidy might not be the main factor driving fermentative capacity vigour in de novo lager hybrids.'

Independant comments on figures:

Figure S2B, similar to Figure 2D, appears to show two groups, and the correlation may be influenced by this artifact.

R: We have removed Figure S2B

I believe there is an issue with Figure S4. Some points appear to have been swapped between the grids in each graph (in A and B). For example, Bioethanol in A does not have 4n, but DBVPG6876 in B (the strain in yellow from bioethanol) has a 4n. The same issue seems to affect Wine; I think Bioethanol and Wine (as well as their strains) are misplaced in some way.

R: We thank the reviewer for carefully pointing out this mistake. We have corrected this Figure by correctly assigning the lineage colour to each strain.

Style:

L70: missing a "." after the references: "consumers attention (Jaeger et al., 2020; Nieto-Villegas et al., 2024)"

R: Incorporated

L72: missing a space before referencs: "and fatty acid esters, in beer and other fermented beverages(Krogerus et al., 2016; Mertens et al., 2015; Pérez et al., 2022; Turgeon et al., 2021)".

R: Corrected

L130: Reference has different font: "and 30 mg/L lysine (Zaret & Sherman, 1985)."

R: Corrected

L164: Reference has different font: "Growthcurver v 0.3.1 (Sprouffske & Wagner, 2016)."

R: Corrected

Reviewer #2 (Comments for the Author):

The manuscript by Zavaleta et al., reports the investigation of hybridization success and the impact of hybridization on fermentation performance and volatile compound profiles in newly formed lager hybrids. The authors observed that the hybrids between *S. cerevisiae* and *S. eubayanus* displayed phenotypic variability, notably influenced by maltotriose consumption, and that triploids exhibited greater phenotypic variability. They further showed that while the *S. cerevisiae* parental lineages primarily influenced aroma production, Beer hybrid lineages mostly resembled the corresponding *S. eubayanus* parent. The transcriptome data suggested the dominant inheritance of the *S. eubayanus* aroma profile by the over-expression of genes related to alcohol metabolism and acetate synthesis in hybrids. Based on convincing data generated by systematic analysis, the present study presented interesting information on the complex interactions between parental lineages and hybridization outcomes, which can be usefully applied for creating yeasts with distinct brewing traits through hybridization strategies. There are a few comments to be addressed with additional information and validation data, as followings, to improve the manuscript in more complete way.

R: We would like to thank the reviewer for providing attention and feedback to our manuscript

Major comment

It is recommended to validate transcriptome analysis data of the polyploid hybrids, HB6 and HB41, by qRT-PCR analysis of a set of DEGs with the primers specific to each parental gene. The lineage specific gene analysis, which can distinguish the genes from *S. eubayanus* or from *S. cerevisiae*, would provide more advanced information on the origin of the up-regulated genes and on the different expression patterns before and after hybridization.

R: We agree that allele-specific expression would be a nice strategy to assess. However, we decided not to take this direction since ploidy differences between parental species might impact the conclusions of this assay. Instead, we decided to take a conservative approximation by comparing orthologs between Hybrids and identifying their expression differences rather than between species. In the future, we will design an experimental strategy to address the reviewer's comment properly.

Minor comments

1). Please provide information on the source of standard volatile compounds used for quantitation in Material and Methods.

R: We have now provided this information.

2) Please check the list of references to unify the format of presentations.

R: We have standardized the font format and followed the mSystems guidelines.

Reviewer #3 (Comments for the Author):

The paper entitled "Understanding brewing trait inheritance in de novo lager yeast hybrids" by Zavaleta et al. explores the fermentative capacities and characteristics of hybrids between *S.*

S. cerevisiae and *S. eubayanus*. The findings are significant not only from a fundamental perspective on hybridization but also in the applied context of beer product diversification.

While the study is interesting, the manuscript requires revision to improve clarity, objectivity, and engagement, particularly in the sections on volatile compounds, RNA-Seq, and the discussion. Specific comments are outlined below:

R: We would like to thank the reviewer for the constructive feedback that improved the clarity of our manuscript. We have addressed all their comments, and we hope that this new version represents a better manuscript version.

Referring to "Beer" lineages can be misleading, as it may be interpreted as referring to the Beer 1 lineage. Since this is not the case, the authors should clarify by using "mixed beer" (at least upon first mention) or another term that avoids confusion while maintaining consistency with the population names used in the original papers.

R: We have incorporated a sentence where we explain the nomenclature (lines 244-245). This sentence reads:

'In contrast, S. cerevisiae strains showed significant differences in fermentative capacity (p-value < 0.05, one-way ANOVA, Figure 1B). Strains from the Mosaic-Beer (hereinafter referred to as 'Beer'), Sake, Bioethanol, and specific strains from the Wine lineages displayed the highest fermentative capacity, with no significant differences compared to W34/70 (Figure 1B).'

Line 43: You want to say phenotypes instead of "genotypes"?

R: We deleted the word 'genotype'.

Line 45: Maybe replace "...lead to replace parental species..." by 'lead to parental replacement...'

R: Replaced.

Line 47: "anthropogenic environmental changes" or anthropogenic transitions?

R: We have amended this sentence for clarity. This now reads:

'Notably, changes in the hybridization frequency between sympatric species have been correlated with novel environmental conditions provided by humans.'

Line 57-58: Needs revision.

R: We have re-written this sentence. This now reads:

'The most well-known example is Saccharomyces pastorianus, the yeast responsible for producing lager-pilsner beer, which ferments at low temperatures and is the most widely produced alcoholic beverage in the world.'

Line 63: You need to refer which hybrids. *S. pastorianus* hybrids combine...

R: Incorporated.

Line 145: Why 42 generations? Has this been proven to be enough for genome stabilization?

R: We have now incorporated a reference to justify this selection. Indeed, it has been previously shown that after 7 passages, hybrids genomes are stable after the rare mating hybridization.

Line 162-164: Why use different wavelengths for adjusting the inoculum and to take the measures during the experiment?

R: Corrected. This now reads:

'The Optical Density (OD) for each well was measured at 620 nm every 30 min for 96 h.'

Line 164 - 165: When you mention the average, do you mean the average from the replicates for each strain? Needs clarification.

R: Indeed. We have incorporated a sentence to clarify this, which reads:

'The average Area Under the growth Curve (AUC) from triplicates for each strain was calculated using the R based tool Growthcurver v 0.3.1 (Sprouffske & Wagner, 2016).'

Line 200: In which conditions were the cells growth and at which stage were they harvested for RNA extraction? I think this information should be added here.

R: We have now incorporated a sentence to clarify this, which reads (203-205):

'RNA was obtained and processed after 24 h under 12°B beer wort fermentation (as indicated in the 'Wort fermentations' subsection) at 12°C in triplicates as previously described {Molinet et al, 2024}. Briefly, RNA from hybrids was extracted using E.Z.N.A. Total RNA Kit 1 (Omega Bio-Tek, USA).'

PCA analysis despite being presented in a main figure is not mentioned in the methods. Please add.

R: We have incorporated this information in the statistical analysis section. This reads:

'The principal component analysis (PCA) was performed using the function prcomp from stats package v. 4.3.1 and the ggfortify v. 0.4.16 and ggplot2 v. 3.4.3 packages for extracting, visualizing and interpreting the results.'

Line 235: From Table S2 it seems to be maltotriose consumption. Not 100% as for maltose, but some are above 15%. This sentence needs to be reformulated.

R: We believe this is within the technical error of the HPLC measurements, and we prefer to take a conservative perspective on the S. eubayanus maltotriose consumption. Historically and based on literature, S. eubayanus cannot consume this sugar under fermentation conditions nor grow in this carbon source.

Line 250: "...from each parent..." doesn't make sense in this sentence.

R: Corrected.

Line 261: Instead of "different letters" you mean 'different numbers'? And the numbers don't reflect the number of hybrids obtained? This part of the legend needs to be clarified.

R: This corresponds to the letters on top of panels A and B results. Panel D depicts the number of hybrids obtained per combination. We have incorporated further details in this sentence for clarity:

'Different letters (from a to g) in Panels A and B reflect statistically significant differences between strains with a p-value < 0.05, one-way ANOVA.'

Line 280: I think "most *S. cerevisiae* lineages" is too strong, because only a handful of lineages were tested.

R: We are aware that only a handful of lineages were tested. Still, based on our experimental design, we observed a strong tendency towards our claim. Instead, we have incorporated a sentence in the discussion section indicating that more non-domesticated lineages should be considered to support this claim further. This sentence reads:

*'The volatile compound profile of *S. cerevisiae* Beer hybrids varied depending on the *S. eubayanus* parent, suggesting that the volatile compound machinery of *S. eubayanus* exhibits a dominant inheritance over that of the Beer *S. cerevisiae* strain in shaping these traits. Future studies assessing the impact of a larger set of lineages, including domesticated and non-domesticated *S. cerevisiae* lineages, might provide further evidence on the VC trait inheritance pattern.'*

Line 331: Can this difference be explained by different parental species rather than ploidy?

*R: Our main message is that the parental lineage, in this case the *S. cerevisiae* lineage exerts the most significant effect on fermentation capacity. In this way, is not only the parental species (*S. cerevisiae*), but also the lineage within this species.*

Line 336: In this section are mentioned the differences between hybrids derived from the beer lineage and those from the wine lineage, with the latter performing better in 12% ethanol. This result should be further explored, as it is closely related to the environmental conditions to which these strains are naturally adapted. A discussion on how these environmental adaptations influence the observed ethanol tolerance should be added.

R: We thank the reviewer for highlighting this result. We have now included an additional sentence in the discussion section (lines 558 – 566). This sentence reads:

*'Interestingly, the observed differences in ethanol tolerance between inter-specific hybrids obtained from the *S. cerevisiae* Beer lineage compared to those obtained from the Wine lineage are likely influenced by the specific environmental conditions to which these strains are naturally adapted (Peter et al, 2018). Strains belonging to Beer lineages have been historically selected for fermentation at lower ethanol concentrations and may not develop the same level of ethanol tolerance as Wine lineages (Gallone et al., 2017; Haas et al., 2019). Wine strains are often exposed to higher ethanol concentrations during grape must fermentation, emphasizing that *S. cerevisiae* lineage-specific phenotypic differences shape the phenotypic traits of interspecific yeast hybrids, such as their capacity to tolerate and thrive in ethanol-rich environments.'*

526: Shouldn't be 'strains' instead of "species"?

*R: We refer to species because, while we do not observe differences between lineages within *S. eubayanus*, distinct differences are evident between lineages within *S. cerevisiae*. We have incorporated a sentence to clarify this issue (lines 533-536):*

*'Our results indicate that hybridization success between diploid *S. eubayanus* and *S. cerevisiae* is common and particularly high in individual lineages, depending on the species. While we did not detect differences between *S. eubayanus* lineages, distinct differences were evident between lineages within *S. cerevisiae*.'*

Line 539: Peter et al. 2018 is not the best example to compare here, as this study was not done with hybrids.

R: This was removed.

Line 554: It is worth mentioning here that the *S. cerevisiae* parental strain is also different between these two hybrids, which certainly also may affect.

R: This was incorporated

Line 556: This sentence doesn't relate to your results and may generate confusion as the hybrids you discuss here (HB6 and HB41) don't have wine subgenomes.

R: This was removed

Line 567-569: This part needs to be rewritten. Is confusing and repetitive.

R: We have rephrased this sentence, particularly in the context of aroma production. This sentence reads (lines 587 – 589):

'Previous studies have described that hybrids with higher DNA content produce higher concentrations of flavor-active esters (Krogerus et al., 2016). However, we generally did not identify a significant correlation between the hybrid's ploidy and VCs best parent heterosis.'

Line 608-609: I think is worth mentioning and reinforcing that is mostly influenced by *S. cerevisiae* parental strain.

R: We now specify this in the conclusion. This sentence reads (lines 643 – 645)

*'Instead, we identified a significant relationship between the specific *S. cerevisiae* parental lineage and the volatile compound and fermentative profiles of the hybrids, indicating lineage-specific inheritance of traits that are crucial for the brewing industry.'*

Re: mSystems00762-24R1 (Understanding brewing trait inheritance in de novo lager yeast hybrids)

Dear Dr. Francisco A Cubillos:

Thank you for your thorough revisions. The reviewers and I both appreciate your efforts and feel the current version of the manuscript is a substantial improvement and suitable for publication. Congrats and thanks for choosing mSystems for this very interesting paper!

Your manuscript has been accepted, and I am forwarding it to the ASM production staff for publication. Your paper will first be checked to make sure all elements meet the technical requirements. ASM staff will contact you if anything needs to be revised before copyediting and production can begin. Otherwise, you will be notified when your proofs are ready to be viewed.

Sincerely,
John Gibbons
Editor
mSystems

Reviewer #1 (Comments for the Author):

Despite the complexity of the suggested changes, most of which aimed for a complete overhaul of the figures to incorporate the new nomenclature, the authors have done an exhaustive job addressing all of them. Also, the manuscript is now much clearer and easier to read. I applaud their efforts!

In summary:

- 1) I find the nomenclature much clearer and easier to follow.
- 2) The clusters in the heatmaps are also much clearer now.
- 3) The new model in Figure 6, showing the impact of various differentially expressed genes on volatile compound production, is fantastic!
- 4) I believe that opening the discussion to other species enriches the discussion, and the authors' ability to make the link to the domestication point is excellent.

I provide some minor changes that do not affect my decision, but that could be useful for the final editing:

- 1) There is a formatting error in L355 (" . ."): ...hybrids from the Beer lineage did not display the highest fitness. .
- 2) There is a formatting error in L476 (" . , " and ".."): ...over-expression on pathways such as biosynthesis of secondary metabolites and the biosynthesis of amino acids. , (Figure 6B)..
- 3) I'm not sure if it's necessary for journal editing purposes, but the next change is not presented in yellow: "The metabolic pathways enriched in each hybrid strain, according to KEGG pathways, indicating HB6's emphasis on ribosome activity and sugar metabolism, while HB41's exhibited over-expression on pathways such as biosynthesis of secondary metabolites and the biosynthesis of amino acids."

Reviewer #2 (Comments for the Author):

Although the main issue regarding the validation of transcriptome data raised by this reviewer was not fully addressed, the revised manuscript has been improved with clearer explanations on a number of major and minor points raised by the other reviewers.

Reviewer #3 (Comments for the Author):

I thank the authors for thoroughly revising the paper and addressing my previous comments.

Just a few minor points:

-Line 119: Please add "as" so the sentence reads '...as previously reported.'

-Wavelength discrepancy: There still seems to be a discrepancy in the wavelengths used for adjusting the inoculum and taking measurements during the experiment. Please check lines 152 ("OD 600") and 154 ("620nm") for consistency.

-Maltotriose consumption: I'm still unclear on your response regarding maltotriose consumption. Could you clarify the error margin on your HPLC measurements and specify the initial maltotriose concentration? This information should be clearer in the manuscript, as the values in the table don't make it immediately obvious.